# Effectiveness of COVID-19 vaccines against severe COVID-19 among patients with cancer in Catalonia, Spain

Felippe Lazar Neto [1,2], Núria Mercadé-Besora[3], Berta Raventós[3,4], Laura Pérez-Crespo[3], Gilberto Castro Junior [2], Otavio T. Ranzani [1,5,7] & Talita Duarte-Salles [3,6,7]

Patients with cancer were excluded from pivotal randomized clinical trials of COVID-19 vaccine products, and available observational evidence on vaccine effectiveness (VE) focused mostly on mild, and not severe COVID-19, which is the ultimate goal of vaccination for high-risk groups. Here, using primary care electronic health records from Catalonia, Spain (SIDIAP), we built two large cohorts of vaccinated and matched control cancer patients with a primary vaccination scheme ($n = 184,744$) and a booster ($n = 108,534$). Most patients received a mRNA-based product in primary (76.2%) and booster vaccination (99.9%). Patients had 51.8% (95% CI 40.3%−61.1%) and 58.4% (95% CI 29.3% −75.5%) protection against COVID-19 hospitalization and COVID-19 death respectively after full vaccination (two-doses) and 77.9% (95% CI 69.2%−84.2%) and 80.2% (95% CI 63.0%−89.4%) after booster. Compared to primary vaccination, the booster dose provided higher peak protection during follow-up. Calibration of VE estimates with negative outcomes, and sensitivity analyses with slight different population and COVID-19 outcomes definitions provided similar results. Our results confirm the role of primary and booster COVID-19 vaccination in preventing COVID-19 severe events in patients with cancer and highlight the need for the additional dose in this population.

The coronavirus disease 2019 (COVID-19) has caused millions of deaths worldwide since the first reported suspected case in Wuhan, China, in late December 2019[1]. Even though most patients will report mild symptoms, nearly 5% will present the severe form of the disease, requiring intensive support[2]. Vulnerable groups at increased risk of severe illness include patients of older age, nursing home facility residents, and those with severe comorbidities, particularly cancer[3]. Compared to healthier individuals, patients with cancer have an

increased risk of death following infection, with an additional incremental risk among those with lung cancer, hematological cancer, or under systemic oncological treatment[4–6]. For instance, in patients with lung cancer, COVID-19-associated mortality was two times higher than in patients without cancer[7].

Randomized clinical trials of COVID-19 vaccines have shown high efficacy and safety in preventing severe outcomes[8–10]; however, these trials targeted the general population and included only patients with

[1]Pulmonary Division, Heart Institute (InCor), Hospital das Clinicas HCFMUSP, Faculdade de Medicina, Universidade de São Paulo, São Paulo, Brazil. [2]Serviço de Oncologia Clínica, Instituto do Câncer do Estado de São Paulo (ICESP), Hospital das Clinicas HCFMUSP, Faculdade de Medicina, Universidade de São Paulo, São Paulo, Brazil. [3]Fundació Institut Universitari per a la recerca a l'Atenció Primària de Salut Jordi Gol i Gurina (IDIAPJGol), Barcelona, Spain. [4]Universitat Autònoma de Barcelona, Bellaterra (Cerdanyola del Vallès), Spain. [5]ISGlobal, Hospital Clínic-Universitat de Barcelona, Barcelona, Spain. [6]Department of Medical Informatics, Erasmus University Medical Center, Rotterdam, The Netherlands. [7]These authors jointly supervised this work: Ranzani Otavio T, Duarte-Salles Talita. ✉e-mail: otavio.ranzani@isglobal.org; tduarte@idiapjgol.org

pre-existing, stable cancers[11], limiting the generalizability of these results to patients with active cancer. Prospective data on immunogenicity following initial vaccination have shown that patients with cancer develop protective antibodies, but in a lower proportion when compared to the general population. This has been observed particularly after the administration of only one vaccine dose among patients with hematological neoplasms, and those undergoing cytotoxic treatments[12–17]. Booster dose administration can elicit strong and durable immune responses in approximately 50% of the patients who were seronegative after the first dose[18–20].

Small retrospective, real-world studies focused on patients with cancer have found low rates of COVID-19 infection following vaccination, and decreased risk of severe disease after infection[21–26]. Cohorts including only vaccinated people have shown that patients with cancer are more susceptible to breakthrough infections compared to healthy individuals[5,27,28], particularly those with only one dose[27,28]. A UK population-level study indicates a reduced temporal cancer COVID-19 fatality rate after vaccination, though still higher than in healthy individuals[29]. Few studies estimated vaccine effectiveness among patients with cancer[30–34] and were either focused on COVID-19 infection[31,32,35] or included cancer as a subgroup of comorbidities[33,34]. Only a minority of them estimated booster vaccine effectiveness[32,33]. Although there is promising evidence on COVID-19 vaccine effectiveness in the general population, we still lack evidence for cancer patients, particularly for booster doses and severe disease.

The combination of large amounts of population-level data and the causal inference framework of target trials[36] can provide valuable opportunities to assess the real-world effectiveness of medical interventions[37] when randomized data is not available. We aim to investigate COVID-19 vaccine effectiveness against severe COVID-19 outcomes among adults with cancer living in Catalonia, evaluating the vaccine effectiveness (VE) of primary schemes against unvaccinated individuals and the relative VE (rVE) of the booster dose compared with two doses.

## Results

### Vaccine uptake
Of 171,284 patients with cancer diagnosis excluding non-melanoma skin cancer between 27th December 2015 and 27th December 2020, 111,576 patients remained after excluding those that died (any cause, N = 41461), moved out from SIDIAP area (N = 3575), had previous COVID-19 (N = 11992), or were nursing home residents (N = 2680) before the beginning of the vaccination campaign. The proportion that received one, two, and three (booster) doses of COVID-19 vaccines were 87.2%, 84.9%, and 68.2%, respectively. Among vaccinated patients, nearly 76% received an initial two-dose mRNA-based vaccination scheme compared to 15% that received two doses of ChAdOX1 and 3.2% that received Ad26.COV2.S as the first dose. The booster dose was composed of almost only mRNA vaccines (76% mRNA-1273 and 24% mRNA-BNT-162b). Suppl. Figure 1 shows the number of COVID-19 cases (Suppl. Figure 1A), the predominant variant of concern (VoC) during each period (Suppl. Figure 1B), and the cumulative vaccine rollout (Suppl. Figure 1C). Suppl. Figure 2 shows the product types and doses of vaccines administered by age groups. The ChAdOX1 vaccine scheme was predominantly administered in adult patients aged 69 years or lower.

### Baseline cohort characteristics
We built two matched cohorts: 184,744 patients (92,372 matched pairs) were included in the first and second dose (primary) vaccination cohort (Cohort A, Suppl. Figure 3) and 108,534 (54,267 matched pairs) in the booster vaccination cohort (Cohort B, Suppl. Figure 4). Compared to un-matched but eligible patients, matched patients had lower proportions of very old (>80 years) and younger (<50 years) patients, a higher proportion of recently diagnosed patients, fewer

comorbidities as per the Charlson Comorbidity Index, fewer diagnoses of metastatic disease, and a higher number of outpatient visits (Suppl. Table 1 and Supp. Table 2).

Matched patients in both cohorts (A and B) had well-balanced characteristics between vaccinated and control groups (Table 1). The majority of patients were older (greater than 60 years old), with a similar proportion of males and females. As expected, the majority of vaccinated individuals in Cohort A received a mRNA-based combination scheme. Among those in Cohort B, approximately 24% previously received the ChAdOx1 combination scheme. The most prevalent cancer diagnosis was breast, followed by prostate and colorectal cancers. 14% and 11% of patients had metastatic disease in Cohort A and B, respectively.

### Vaccine effectiveness
Figure 1 shows the cumulative incidence for the primary outcome of COVID-19-associated hospitalization, and Fig. 2 and Fig. 3 shows the estimated VE for each time period for the primary vaccination (Cohort A) and booster vaccination (Cohort B) cohorts respectively. For the primary vaccination, the estimated VE against COVID-19 hospitalization for the first (partially vaccinated) and second dose (fully vaccinated) was 42.0% (95%CI 22.3 - 56.7) and 51.8% (95%CI 40.3 - 61.1) respectively. When expanding the time periods, we observed an increase in VEs, particularly after the second dose, which peaked after 60 days (58.4%, 95%CI 34.5 - 73.6) but waned after 120 days (-19.7%, 95% CI -52.7 - 26.6). For the booster vaccine, we found high rVE (> 75%) already in the immediate period post-vaccination, which remained high until 120 days, when a decrease in rVE was observed. Visual inspection of cumulative incidence graphs during the initial period of 0 to 14 days after vaccination (Suppl. Figure 5) shows low residual confounding for both cohorts. Results for the secondary outcomes COVID-19 severe hospitalization, and COVID-19 deaths showed similar results (Fig. 2 and Fig. 3). Competing hazards model (all-cause death as competing risk) showed comparable results (Suppl. Table 3).

### Subgroup analysis
Figure 4 shows the results for subgroup effect modification analysis for both cohorts. For the primary vaccination scheme (Cohort A), subgroup analysis has found lower VE after full vaccination for the elderly (33.2% vs 74.7%, $p < 0.001$, Suppl. Figure 6) and metastatic patients (24.0% vs 59.6%, $p = 0.025$), and no effectiveness during the Delta VoC. For the booster (Cohort B), we found higher VE after 14 days for older (82.2% vs 49.3%, $p = 0.028$, Suppl. Figure 7), and male patients (81.4% vs 69.8%, p = 0.006). We found a numerical increase in rVE for those who previously did not receive the mRNA-based vaccination scheme (between 14 and 60 days 80.5% vs 77.4%, after 60 days 73.7% vs 39.4%, $p = 0.155$). We did not find any effect modification by a diagnosis of hematological malignancy and a lower vaccine effectiveness during the Omicron period for the booster dose compared with the Delta period.

### Negative Outcomes Calibration
Negative outcomes estimands for each cohort and period are shown in Suppl. Figures 8 and 9. After adjustment for negative outcomes, VE against COVID-19 hospitalization for the primary vaccination scheme was 42.7% (95% CI 11.2 - 63.0) for partially vaccinated and 58.2% (95% CI 43.8 - 68.9) for fully vaccinated individuals. For the booster dose, calibrated VE was 73.5% (95%CI 61.3 - 81.8) during the 14 - 60 days period and 50.8% (95% CI -1.2 - 76.0) after 60 days (Suppl. Table 4). Sensitivity analysis with an expanded set of negative outcomes provided similar results (Suppl. Table 4).

### Non-COVID outcomes
Vaccination was associated with decreased hazards of all-cause hospitalizations and non-COVID deaths in the immediate period post-

**Table 1 | Baseline characteristics of patients from the primary vaccination cohort (cohort A) and booster vaccination cohort (cohort B)**

| | 1st and 2nd Dose Cohort (Cohort A) | | Booster Cohort (Cohort B) | |
|---|---|---|---|---|
| | Unvaccinated | Vaccinated | Two Doses | 3rd dose |
| Number of Patients | 92372 | 92372 | 54267 | 54267 |
| Age, Mean (SD) | 64.80 (15.12) | 64.89 (15.14) | 69.74 (12.43) | 69.74 (12.42) |
| Age Group (years), N (%) | | | | |
| 18-49 | 15521 (16.8) | 15521 (16.8) | 3631 (6.7) | 3631 (6.7) |
| 50-59 | 15805 (17.1) | 15805 (17.1) | 8024 (14.8) | 8024 (14.8) |
| 60-69 | 22831 (24.7) | 22831 (24.7) | 14332 (26.4) | 14332 (26.4) |
| 70-79 | 23728 (25.7) | 23728 (25.7) | 16521 (30.4) | 16521 (30.4) |
| 80-115 | 14487 (15.7) | 14487 (15.7) | 11759 (21.7) | 11759 (21.7) |
| Female Sex, N(%) | 47017 (50.9) | 47017 (50.9) | 26653 (49.1) | 26653 (49.1) |
| MEDEA deprivation index, N (%) | | | | |
| Missing | 11119 (12.0) | 10459 (11.3) | 5163 (9.5) | 4972 (9.2) |
| Rural | 13519 (14.6) | 13765 (14.9) | 8337 (15.4) | 8430 (15.5) |
| Urban, Quintile 1 | 14099 (15.3) | 15154 (16.4) | 9312 (17.2) | 9469 (17.4) |
| Urban, Quintile 2 | 13920 (15.1) | 14353 (15.5) | 8789 (16.2) | 8841 (16.3) |
| Urban, Quintile 3 | 13600 (14.7) | 13624 (14.7) | 8141 (15.0) | 8164 (15.0) |
| Urban, Quintile 4 | 13723 (14.9) | 13448 (14.6) | 8044 (14.8) | 7946 (14.6) |
| Urban, Quintile 5 | 12392 (13.4) | 11569 (12.5) | 6481 (11.9) | 6445 (11.9) |
| 1st and 2nd Vaccine Combination, N (%) | | | | |
| ChAdOx1-ChAdOx1 | | 16819 (18.2) | 12814 (23.6) | 12814 (23.6) |
| mRNA-1273-mRNA-1273 | | 16946 (18.3) | 5037 (9.3) | 5037 (9.3) |
| mRNA-BNT162b-mRNA-BNT162b | | 53466 (57.9) | 36416 (67.1) | 36416 (67.1) |
| Ad26 Only | | 2967 (3.2) | | |
| 3rd Vaccine, N (%) | | | | |
| Pfizer-mRNA-BNT162b | | | | 13649 (25.2) |
| Moderna-mRNA-1273 | | | | 40613 (74.8) |
| AZ-ChAdOx1 | | | | 5 (0.0) |
| Ad26 | | | | |
| Time Since Diagnosis (Years), N (%) | | | | |
| Less than one | 23985 (26.0) | 23985 (26.0) | 11604 (21.4) | 11604 (21.4) |
| One | 18775 (20.3) | 18775 (20.3) | 11263 (20.8) | 11263 (20.8) |
| Two | 17554 (19.0) | 17554 (19.0) | 10869 (20.0) | 10869 (20.0) |
| Three | 16285 (17.6) | 16285 (17.6) | 10254 (18.9) | 10254 (18.9) |
| Four | 15773 (17.1) | 15773 (17.1) | 10277 (18.9) | 10277 (18.9) |
| Cancer Diagnosis, N (%) | | | | |
| Breast | 15159 (16.4) | 15690 (17.0) | 8911 (16.4) | 9137 (16.8) |
| Prostate | 11259 (12.2) | 11338 (12.3) | 7810 (14.4) | 8036 (14.8) |
| Colorectal | 12074 (13.1) | 12125 (13.1) | 7687 (14.2) | 7706 (14.2) |
| Lung | 5491 (5.9) | 5198 (5.6) | 2717 (5.0) | 2509 (4.6) |
| Head and Neck | 2827 (3.1) | 2703 (2.9) | 1617 (3.0) | 1539 (2.8) |
| Endometrium | 2041 (2.2) | 1929 (2.1) | 1299 (2.4) | 1182 (2.2) |
| Cervix/Uterus | 2142 (2.3) | 1835 (2.0) | 1070 (2.0) | 971 (1.8) |
| Bladder | 8243 (8.9) | 8209 (8.9) | 5654 (10.4) | 5453 (10.0) |
| Biliary/HCC | 1820 (2.0) | 1653 (1.8) | 808 (1.5) | 791 (1.5) |
| Melanoma | 3460 (3.7) | 3597 (3.9) | 1984 (3.7) | 2219 (4.1) |
| Pancreas | 1456 (1.6) | 1342 (1.5) | 654 (1.2) | 576 (1.1) |
| Kidney | 3279 (3.5) | 3382 (3.7) | 1938 (3.6) | 2064 (3.8) |
| Gastric | 1657 (1.8) | 1502 (1.6) | 860 (1.6) | 817 (1.5) |
| Esophagus | 518 (0.6) | 477 (0.5) | 219 (0.4) | 224 (0.4) |
| Testis | 624 (0.7) | 607 (0.7) | 197 (0.4) | 186 (0.3) |
| Thyroid | 1778 (1.9) | 1667 (1.8) | 852 (1.6) | 828 (1.5) |
| Central Nervous System | 902 (1.0) | 912 (1.0) | 316 (0.6) | 301 (0.6) |
| Neuroendocrine Cancers | 394 (0.4) | 429 (0.5) | 245 (0.5) | 230 (0.4) |
| Sarcomas | 968 (1.0) | 920 (1.0) | 443 (0.8) | 461 (0.8) |

**Table 1 (continued) | Baseline characteristics of patients from the primary vaccination cohort (cohort A) and booster vaccination cohort (cohort B)**

|  | 1st and 2nd Dose Cohort (Cohort A) | | Booster Cohort (Cohort B) | |
|---|---|---|---|---|
| Leukemia | 2270 (2.5) | 2431 (2.6) | 1244 (2.3) | 1259 (2.3) |
| Multiple myeloma | 876 (0.9) | 1131 (1.2) | 538 (1.0) | 462 (0.9) |
| Lymphoma | 2970 (3.2) | 3170 (3.4) | 1544 (2.8) | 1437 (2.6) |
| Hematological | 2186 (2.4) | 2119 (2.3) | 3310 (6.1) | 3106 (5.7) |
| Other | 7877 (8.5) | 7889 (8.5) | 4168 (7.7) | 4122 (7.6) |
| Charlson Comorbidity Index, Median [IQR] | 3 [2, 6.00] | 3 [2, 6] | 3 [2, 5] | 3 [2, 5] |
| Metastatic Solid Tumor, N (%) | 13852 (15.0) | 13507 (14.6) | 6399 (11.8) | 5952 (11.0) |
| Number of Outpatient Visits*, N (%) | | | | |
| 0 | 11333 (12.3) | 9029 (9.8) | 4982 (9.2) | 4380 (8.1) |
| 1 | 8834 (9.6) | 8998 (9.7) | 5208 (9.6) | 5137 (9.5) |
| 2 | 8632 (9.3) | 8694 (9.4) | 5157 (9.5) | 5025 (9.3) |
| ≥ 3 | 63573 (68.8) | 65651 (71.1) | 38920 (71.7) | 39725 (73.2) |

*During the year prior to vaccination campaign beginning (27th December 2020).

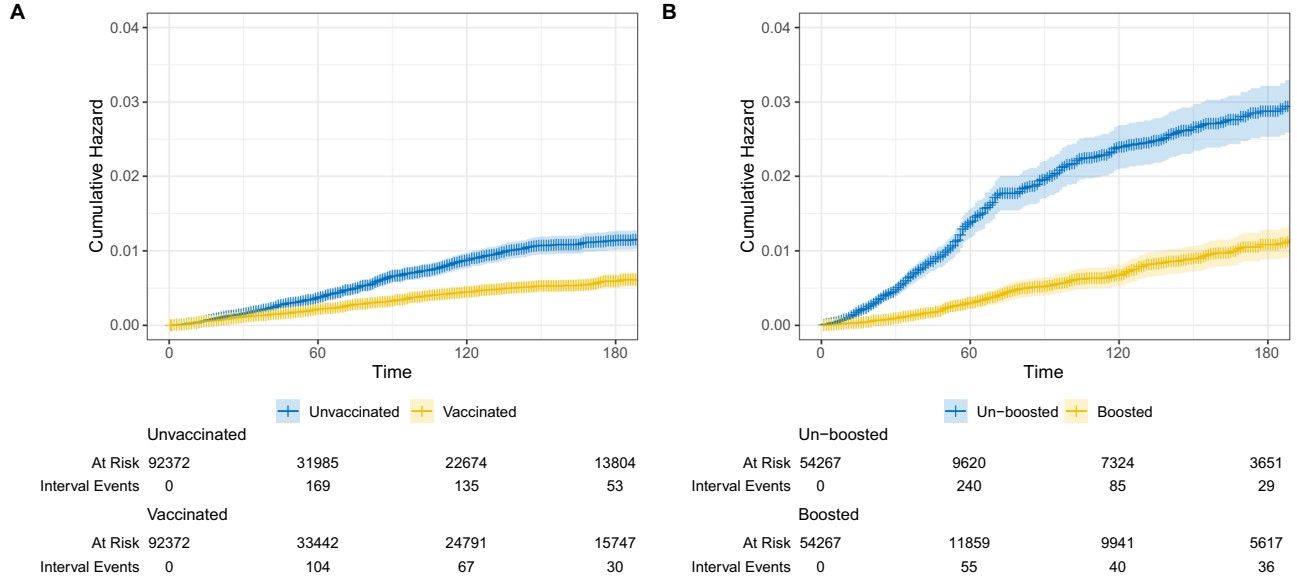

**Fig. 1 | Incidence of COVID-19 hospitalizations.** Cumulative incidence of COVID-19 hospitalizations (primary outcome) for the primary vaccination (Figure **A**) and booster vaccination (Figure **B**) cohorts between vaccinated and control groups. The solid lines represent the estimated cumulative hazards, while the shaded areas indicate the 95% confidence intervals.

vaccination for both cohorts (Suppl. Tables 5 and 6). For Cohort A, primary vaccination was associated with a non-significant decrease in all-cause hospitalizations during follow-up but a sustained decreased hazard for non-COVID death (Suppl. Table 5). For Cohort B, booster vaccination was associated with a lower risk of all-cause hospitalizations and non-COVID death in all time periods (Suppl. Table 6). We observed a higher proportion of non-COVID-19 deaths without preceding hospitalization than COVID-19 deaths (Suppl. Table 7). Analysis of health services usage (outpatient, telehealth, home, inpatient, and ICU visits) by vaccination status showed lower likelihood in the vaccination group for most outcomes in cohort B, but not for Cohort A; which showed increased hazards for tele-health, home and outpatient visits and lower hazards for inpatient and ICU visits (Suppl. Table 8).

**Sensitivity analysis**

Sensitivity analysis including previous influenza vaccine receipt in matching and excluding those hospitalized a month prior to

vaccination in the primary and booster cohorts (restricted matching cohort, Suppl. Tables 9 and 10) reduced the vaccine protection for all-cause hospitalizations, but not non-COVID death, which remained lower in the vaccinated group (Suppl. Tables 11 and 12, Suppl. Figures 10 and 11). The COVID-19 primary outcome had comparable results (Suppl. Table 13).

Additional sensitivity analyses with different definitions for the cancer cohort and COVID-19 outcomes showed little deviations from the original results (Suppl. Figures 12 and 13).

## Discussion

In this matched cohort study, we found that the two-dose primary vaccination scheme effectively reduced COVID-19 hospitalization and mortality outcomes in individuals with cancer, and that administering a booster dose provided substantial and meaningful additional protection for those who had already received the initial two-dose vaccination scheme. We found that the booster dose provided higher peak effectiveness over time.

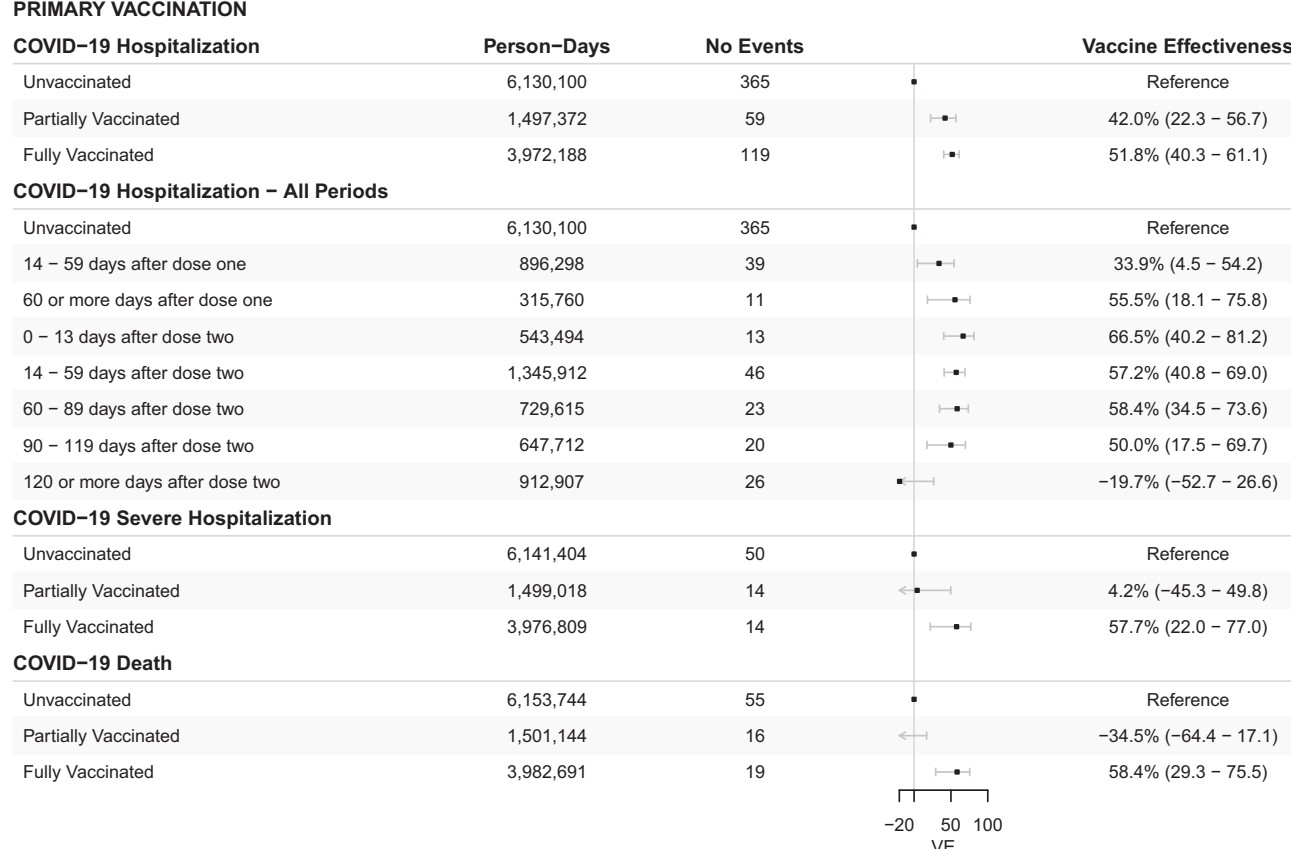

**Fig. 2 | Vaccine effectiveness for first and second doses.** Forest plot of estimated vaccine effectiveness (point estimate) and its 95% confidence interval (error bars) since the time from vaccination for the initial vaccination scheme (Cohort A). Here we show vaccine effectiveness for the primary (hospitalization) and secondary endpoints (severe hospitalization, and death) separately. VE = Vaccine Effectiveness.

Two test-negative case-control studies using a network of hospitals across nine states in the United States[34] and linked cancer registry with surveillance data in the United Kingdom (UK)[31] have estimated the initial two-dose VE for hospitalization among patients with cancer as 79% (95% CI, 73-84%) and 84.5% (95% CI, 83.6-85.4) respectively, which is slightly higher than what we found in our study: 52% after the second dose. Regarding the first booster, another study from the same group in the UK[32] showed a VE of 80.5% (95% CI, 77.3-83.2) against hospitalization when comparing booster vs. unvaccinated population, which is not straightforward to compare with our estimate of relative effectiveness (i.e., booster compared with primary vaccination scheme protection). There are several possible explanations for the observed differences, including characteristics both at the individual level, such as different population distribution on age, sex, income, cancer staging, healthy-seeking behavior, vaccine type, and time of follow-up, and at the local level, such as pandemic period and predominant VoC during effectiveness evaluation.

Although previous investigations[30–32] have indicated decreased VE for COVID-19 infection among patients with a hematological neoplasm or within one year of initial diagnosis, we did not find decreased effectiveness for severe COVID-19 for either of these subgroups. This is likely explained by the different outcomes we investigated: severe disease (COVID-19 hospitalization or death), instead of mild COVID-19 infection. For example, patients with lymphoma in the UK cohort had a 10.5% reduction in breakthrough infections following booster dose but a 80% reduction in COVID-19 death[32]. The lack of subgroup analysis for severe outcomes in previous studies limits further comparisons[30–32].

Older patients (≥ 65 years) had lower VE following the initial scheme, as previously described for the general population[38,39].

Interestingly, booster vaccination among this subgroup provided additional significant protection, better explained by the probable different baseline risk following initial vaccination and the higher relative VE estimated. Serological studies have shown that approximately half of seronegative oncological patients can have seroconversion following a booster administration. We found a similar effect for sex: male patients had possibly lower VE following the initial scheme and higher after the booster. Past studies have shown that among vaccinated individuals, male patients may still have a higher risk of severe COVID-19 outcomes[27,40], and serological studies suggest a higher waning effect compared to females[27,41], which can help explain baseline risk differences and different relative effectiveness after booster vaccination.

We found a waning effect for the initial two-dose vaccination scheme but a higher peak effectiveness and more sustained protection against COVID-19 death for the booster dose (> 70% two months after booster). In addition, we have shown that the booster dose was likely effective during the Omicron period (rVE 29.3% 95%CI -1.4 - 50.7), in line with previous serological studies in the general population showing maintained protection against Omicron for mRNA vaccines[42]. This is one of the first studies to assess waning in both schemes (two-dose initial scheme and booster) and evaluate the impact of the VoC period on effectiveness in patients with cancer. Although results are promising, they should be interpreted having in mind the potential bias introduced by susceptibles depletion[43] and undocumented infections. We found an increased numerical benefit for those who received a mRNA booster after a two-dose ChAdOX1 homologous initial scheme. Previous evidence suggests that heterologous schemes may provide additional protection compared to homologous

**BOOSTER VACCINATION**

| COVID–19 Hospitalization | Person–Days | No Events | | rVE |
|---|---|---|---|---|
| Un–boosted | 2,230,862 | 360 | | Reference |
| 14 – 59 days after booster | 786,371 | 43 | | 77.9% (69.2 – 84.2) |
| 60 or more days after booster | 1,281,426 | 88 | | 45.8% (28.7 – 58.9) |
| **COVID–19 Hospitalization – All Periods** | | | | |
| Un–boosted | 2,230,862 | 360 | | Reference |
| 14 – 27 days after booster | 329,755 | 12 | | 81.0% (64.8 – 89.8) |
| 28 – 59 days after booster | 456,616 | 31 | | 76.4% (64.9 – 84.1) |
| 60 – 119 days after booster | 638,777 | 40 | | 63.6% (47.1 – 75.0) |
| 120 or more days after booster | 642,649 | 48 | | 5.3% (−31.7 – 38.7) |
| **COVID–19 Severe Hospitalization** | | | | |
| Un–boosted | 2,239,437 | 31 | | Reference |
| 14 – 59 days after booster | 786,889 | < 5 | | 87.0% (42.9 – 97.1) |
| 60 or more days after booster | 1,283,441 | 6 | | 59.2% (−9.3 – 84.9) |
| **COVID–19 Death** | | | | |
| Un–boosted | 2,263,411 | 105 | | Reference |
| 14 – 59 days after booster | 791,359 | 12 | | 80.2% (63.0 – 89.4) |
| 60 or more days after booster | 1,290,609 | 16 | | 72.2% (50.5 – 84.4) |

rVE scale: −20 0 50 100

**Fig. 3 | Relative vaccine effectiveness of booster dose.** Forest plot of estimated COVID–19 relative vaccine effectiveness (point estimate) and its 95% confidence interval (error bars) since the time from vaccination for the booster vaccination (Cohort B). Here we show vaccine effectiveness for the primary (hospitalization) and secondary endpoints (severe hospitalization, and death) separately. rVE = Relative Vaccine Effectiveness. Counts below five have been masked to protect patients' privacy.

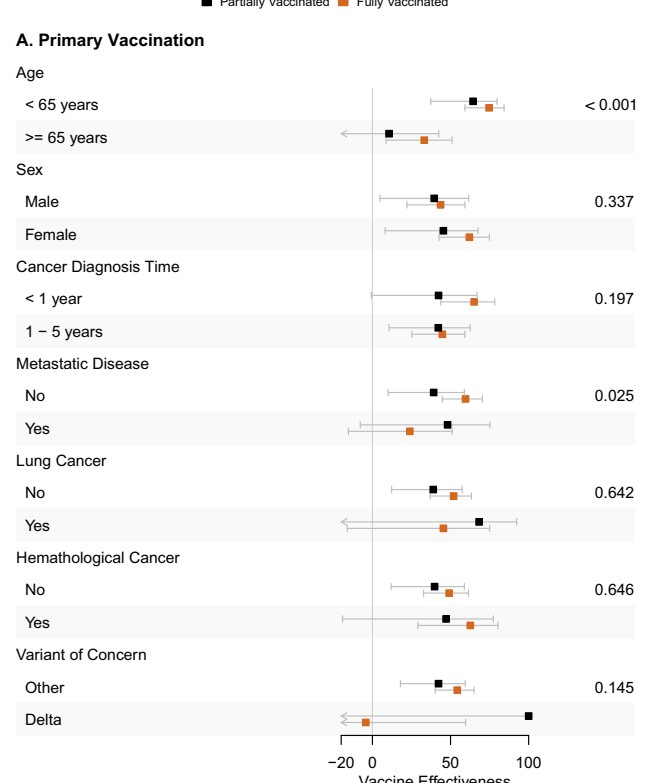

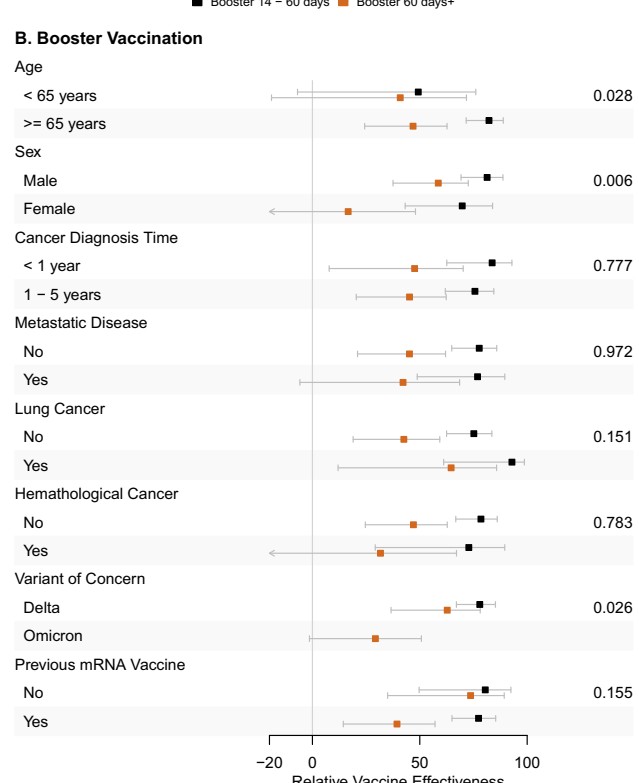

**Fig. 4 | Vaccine protection among subgroups of interest.** Forest plot of estimated COVID–19 vaccine effectiveness (point estimate) and its 95% confidence interval (error-bars) among subgroups for the primary vaccination (Cohort A) and booster vaccination (Cohort B). The detailed number of events, observations and precise confidence interval estimates for each sub-group are shown in Suppl. Figures 6 and 7.

schemes[44]. Because of the restricted range of ages (mostly ≤ 69 years) that received initial ChAdOX1 vaccination, results might be not generalizable to all adults with cancer, particularly the older ones that did not receive ChAdOX1.

Estimation of VE from observational studies[45] is challenging as vaccinated patients are often healthier (healthy bias) and more health conscious (health-seeking bias) than unvaccinated individuals. Although proper matching on relevant covariates may provide well-balanced characteristics between groups, it is often insufficient to adjust for unmeasured confounders. Previous research on COVID-19 vaccine effectiveness has shown that, despite proper matching, vaccinated patients had a lower risk of non-COVID death compared to control[45–47], partially explained by healthier conditions of vaccinated patients. In our study, we showed a lower risk of non-COVID deaths among vaccinated patients but a much lower magnitude for all-cause hospitalizations. Further adjustment for previous influenza receipt and exclusion of recently hospitalized patients, variables that would capture part of the healthy and health-seeking biases, improved all cause-hospitalization differences between groups, but not non-COVID death. We hypothesize two other reasons for this finding. First, the principal difference between vaccinated and control ones occurred in the immediate period after the vaccine (Suppl. Table 5 and 6), a finding observed by other studies and improbable to be related to the biological action of vaccines. This is likely because the most ill or frail individuals ended up dying early on after matching, and in a clinical trial, they would likely not be eligible to be randomized. Second, part of the non-COVID-19 deaths could be actually COVID-19 deaths, leading to misclassification of the outcome. In the analyzed population, there is a high proportion of non-COVID-19 deaths that occurred without hospitalization (60% for Cohort A and 71% for Cohort B), which is associated with a great network of home and palliative care in the region, making us hypothesize that these individuals are unlikely to get tested (and consequently being diagnosed)[48]. Finally, socio-economic variables in large databases can not capture all socio-economic nuances and wealthy bias is a concern. The comparable risk in all-cause hospitalizations and similar effectiveness after negative outcomes calibration and across the sensitivity analyses result in confidence in the vaccine protection observed. However, the magnitude and direction of the potential bias is uncertain.

This study has many strengths. We included a large number of patients in each assessment - 184,744 patients for the two-dose initial vaccination and 108,534 patients for the booster analysis. In addition, cancer diagnosis had been previously validated in the SIDIAP database with good agreement with population-based cancer registries in Catalonia[49] providing a quality assessment of our population of interest. We designed an observational study with robust methodology including a target-trial framework with rolling entry matching on a daily basis with adjustment for all potential observable confounders and similar baseline characteristics between groups. This methodology has already been validated in similar scenarios of vaccine effectiveness[38,50] with valid estimates of effectiveness. We calibrated our estimates with negative control outcomes that are highly improbable to be associated with our exposure of interest, addressing unobservable residual confounding which is a major issue in large population studies. Finally, we provided a comprehensive analysis of our findings, presenting sensitivity analyses with different outcomes and cohort definitions, including non-COVID outcomes, and creating additional cohorts matched on previous influenza receipt to investigate for potential healthy-vaccine bias.

The main limitation of this study is its observational design. Although large observational databases with adequate methodology may duplicate the results of randomized clinical trials[37], residual confounding cannot be excluded. However, we tried to minimize the chances of confounding by including variables associated with health-seeking behaviors (number of outpatient visits) and socioeconomic factors (the MEDEA deprivation index) in addition to calibrating for negative control outcomes. Visual inspection of bias indicators (i.e., during the initial days following vaccination)[51,52] showed a low risk of bias, and negative control adjustment showed similar results. Although the selection of negative outcomes might be debatable, we have chosen negative outcomes previously validated in very similar settings[53]. However, the decreased hazards of non-COVID death, particularly in the immediate period post-vaccination, indicates residual healthy bias. We attempt to reduce this bias by matching new cohorts on previous influenza vaccine receipt, excluding frail patients (patients hospitalized a month prior), and adjusting variables that could capture healthy-seeking behavior, but lower hazards for non-COVID death persisted, and our results should be interpreted in light of these findings. It is possible that the VE and rVE estimates are overestimated because of this residual healthy bias, particularly in the immediate period post-vaccination when most non-COVID deaths were observed. We defined COVID-19 outcomes based on the temporal association of a positive diagnosis and the outcome (hospitalization and death) as causes for hospitalization and death were unavailable; however, sensitivity analysis with different definitions provided similar results. Another limitation is the lack of data granularity regarding cancer staging and treatment (including chemotherapy and radiation therapy, among others), which limits our capability of answering questions regarding the timing and type of treatment provided. To overcome this limitation, we used time from cancer diagnosis as a surrogate for cancer treatment receipt, as patients with recent diagnoses may have higher chances of being under treatment. Additionally, during the COVID-19 pandemic, patients at higher risk of severe outcomes might have taken additional measures to prevent infection, such as avoiding gatherings or using face shields which are not captured by data and may influence results[54]. Lastly, during the Omicron wave, we could not differentiate between patients who have been hospitalized with COVID-19 and not because of COVID-19. However, this is even harder to ascertain in patients with cancer. We included a sensitivity analysis with COVID-19 diagnosis up to 3 days after admission with similar results, showing that the time of COVID-19 infection (pre-admission vs during admission) did not change outcomes.

In conclusion, we found a significant protective effect for both the primary and booster vaccination schemes on hospitalization and mortality outcomes among patients with cancer. The booster protective effect was high and more durable, particularly against COVID-19 death. Patients should be encouraged to get vaccinated if not and boosted if they have had only two doses. Because of the higher risk of breakthrough infections, hospitalizations, and death compared to healthy individuals, patients with cancer should be prioritized in future additional dose studies and vaccination campaigns.

## Methods

### Study design, settings, and data source

We conducted a matched population-based cohort study using the Information System for Research in Primary Care (SIDIAP; www.sidiap.org) database[55]. SIDIAP is a primary care longitudinal database from Catalonia, Spain, which contains pseudo-anonymized individual-level patient data since 2006, with 5.8 million people active in June 2021 (75% of the Catalan population). The present study was conducted using data from December 27th, 2020 to June 30th, 2022. The SIDIAP database includes clinical diagnosis, lifestyle information, and dispensed medications in primary care, including COVID-19 vaccine products, linked to both the SARS-CoV-2 polymerase chain reaction (RT-PCR) and rapid antigen tests results database and hospital records database. The SIDIAP has been mapped to the Observational Medical Outcomes Partnership (OMOP) Common Data Model (CDM), allowing the reproducibility of study definitions across a wide range of mapped databases[56,57]. The current work was approved by the Clinical Research Ethics Committee of IDIAPJGol (project code 23/023-EOm).

The Spanish national COVID-19 vaccination campaign was launched on 27th December 2020. Because of the initially limited availability of vaccines, groups considered at higher risk were prioritized, including healthcare workers, nursing home residents, and older subjects[58]. According to national guidelines, patients with cancer under active treatments or other serious immunosuppressive comorbidities were prioritized before the general population but only after older patients. Vaccine products used for the national COVID-19 vaccination campaign were progressively extended as new authorizations were granted and included: BNT162b2 (*Pfizer*, mRNA, two-dose, 21-day interval), ChAdOx1 (*AstraZeneca*, adenovirus, two-dose, 3-month interval), mRNA-1273 (*Moderna*, mRNA, two-dose, 28-days interval), and Ad26.COV2.S (*Janssen*, adenovirus, single-dose). People who were vaccinated with Ad26.COV2.S were later recommended for an additional second dose because of lower expected vaccine effectiveness at that time[59].

### Study population, exposure definitions, and outcomes

Eligible individuals were adults aged 18 or older with at least one year of prior observation, with a record of cancer diagnosis (excluding non-melanoma skin cancer) within the last 5 years prior to the index date, which represents the date at which individuals were eligible to enter the matching pool. We excluded patients with any previous diagnosis of COVID-19, defined as a combination of either clinical or laboratory diagnoses of COVID-19 prior to the index date. Patients who were transferred out of SIDIAP before matching and those living in nursing homes were also excluded.

The exposure of interest in this study was COVID-19 vaccination defined as the receipt of any COVID-19 vaccine among all available vaccine products at the time (BNT162b2, mRNA-1273, ChAdOx1, and Ad26.COV2.S). Overall vaccine uptake was described including all eligible patients at the beginning of the vaccination campaign on 27th December 2020.

We built two matched cohorts: Cohort A to evaluate the VE of the first and second doses (primary vaccination) compared with unvaccinated individuals; and Cohort B to evaluate the relative VE of the booster compared with two doses. Cohort A included all adults eligible for primary vaccination with a cancer diagnosis up to five years before the first dose vaccination date. Cohort B, a subset of Cohort A, included only patients that previously received homologous vaccine schemes with BNT162b2, mRNA-1273, and ChAdOx1, with adequate interval between the second and third dose (a minimum of 90 days intervals for ChAdOx1 and 180 days for BNT162b2 and mRNA-1273). For Cohort B, patients had to have a cancer diagnosis before the first dose vaccination date up to five years from the booster vaccination date to ensure the patient had a cancer diagnosis at the time of the primary vaccination.

The index date was set to the first dose date for vaccinated individuals in Cohort A and the booster date for exposed individuals in Cohort B. The Index date for patients in the control groups was set as the index date of their matched counterparts. The study end date for Cohort A was on 20th November 2021 (one month before the Omicron VoC predominance) and for Cohort B at the last available information date (30th June 2022)

Predominant variants of concern (VoC) (Delta, Omicron, and others) at each time period were defined as ≥ 50% of weekly tested samples in Catalonia, extracted from the Global Initiative on Sharing All Influenza Data (GISAID)[60].

The primary outcome of this study was COVID-19 hospitalization, and secondary outcomes included COVID-19 severe hospitalization - defined as a COVID-19 hospitalization with the need for invasive oxygen supplementation, and COVID-19 death. Consistent with prior research on severe COVID-19[61,62], we defined COVID-19 hospitalization as any hospital admission within 21 days from the COVID-19 diagnosis up to the entire duration of hospitalization. Likewise, death was

classified as any cause of death occurring within 28 days from the date of COVID-19 diagnosis. In addition to COVID-19-associated outcomes, we report all-cause hospitalizations and non-COVID deaths. A complete list of variables, exposures and outcome definitions can be found in Suppl. Tables 14, 15 and 16. The decision to set COVID-19 hospitalization as primary outcome (in contrast with the composite outcome of COVID-19 hospitalization and/or death) and evaluate COVID-19 severe hospitalization, all-cause hospitalizations and non-COVID-19 deaths was made post-hoc during peer review.

### Statistical analysis

We emulated a pragmatic target trial of COVID-19 vaccination among patients with cancer within the SIDIAP database using rolling entry matching (REM)[45,63], on a daily basis (Suppl. Figure 14). For Cohort A, eligible patients upon their first dose date were matched in a 1:1 ratio to eligible un-vaccinated individuals. For Cohort B, individuals with a completed primary scheme upon their booster dose date were matched in a 1:1 ratio to individuals with a completed primary scheme and eligible to receive a booster dose. Matching was performed by combining exact and caliper matching. Age (bins of five years), sex, cancer diagnosis time (categories zero to five years), the municipality of residence, the MEDEA deprivation index[64] (a proxy for deprivation based on place of residence from Q1 [least deprived] to Q5 [most deprived]), the number of outpatient visits in the previous year (as a proxy for healthy seeking behavior), the Charlson Comorbidity Index and metastatic disease were used to build a propensity score with a logistic regression model, which was used for matching in a caliper of 0.01 while ensuring exact matching for age, sex, cancer diagnosis time and the municipality of residence. For Cohort B, the exact matching also included the previous vaccination scheme (i.e., first and second dose product). Matching variables were chosen based on their potential association with receiving the vaccine (exposure of interest) and the risk of severe COVID-19 (outcome of interest).

After matching, patients were followed until an outcome event of interest occurred, death, or were censored at the last follow-up date. In case the control-matched patient was vaccinated, the matched-pair was censored at the date of control vaccination. Patients who had COVID-19 infection but were not hospitalized nor died were censored 28 days after the date of infection to account for a window of susceptibility to events of interest. The censoring of those who had COVID-19 without an event of interest was based on the following rationale: we are evaluating the first COVID-19 infection-associated event, and those with a previous infection would have low susceptibility to subsequent infections and consequently low risk of the outcome (i.e., an individual is still at risk during its first COVID-19 infection and subsequent window of susceptibility to the event, but this risk decreases if no event occurs and immunity is created).

Baseline covariable balance was evaluated by standardized mean differences (SMD) and descriptive characteristics between groups. Continuous variables were described with mean, median, standard deviation (SD), or interquartile range depending on variable distribution, and categorical variables with absolute numbers and relative proportions.

We calculated and plotted the cumulative incidence between groups with the Kaplan-Meyer estimator. VE was estimated using a Cox Proportional hazards model in the matched cohorts as 1 minus the hazard ratio (HR) between groups. HR and 95% confidence intervals (CI) were calculated in different periods of time (time-stratified Cox model) since vaccination (day zero) to investigate the long-term effectiveness and waning effect. Our primary analysis estimated the VE for the period after 14 days of the first dose (partially vaccinated), after 7 days of the second dose (fully vaccinated), and relative VE 14 days and 60 days after the third dose. Potential vaccine waning was evaluated with a larger number of period breaks from vaccination until 120 days or more thereafter (all periods). All Cox models accounted for

the competing risk of death under the framework of cause-specific competing risks, more suitable for etiological research questions[65–68]. In a post-hoc decision during the peer review, we also estimated the VE using the Fine-Gray models considering the competing risk of death, deriving sub-distribution HRs for the main analysis, an approach more suitable for prediction and prognostic, to complement the cause-specific evaluation[65,69]. The competing event was all-cause death for the COVID-19 hospitalization outcome.

We investigated residual confounding by visually inspecting the cumulative incidence and estimated VE differences in the immediate (0 - 14 days) post-vaccination period when no protection is expected[52]. In addition, residual confounding could remain after matching due to unobserved confounding variables, particularly ones associated with wealthy-healthy bias that may affect results. Thus, we performed a negative control outcomes (NCO) analysis[70] with a previously published list of 43 validated outcomes that were highly improbable to be associated with our exposure of interest (COVID-19 vaccination)[53]. In addition to these previous validated negative outcomes, we performed an additional analysis including 11 additional outcomes in the negative outcomes set (Suppl. Table 17). Results from NCO were used to empirically calibrate our estimates with the EmpiricalCalibration package in R[53,71]. Additionally, we report estimates of health-services-seeking behavior, including outpatient visits, telehealth visits, home visits, and inpatient and ICU visits after vaccination. The decision to evaluate these other negative control outcomes (expanded set of negative outcomes and health-services-seeking behavior) was made post-hoc during peer review.

Subgroup effect modification of the primary outcome was defined based on previous knowledge of possible effect modifiers in this population and included: age (<65 years old vs ≥ 65 years old), sex (male vs female), cancer diagnosis time (1 year vs 1-5 years), metastatic disease (yes vs no), lung cancer diagnosis (yes vs no), hematological cancer diagnosis (yes vs no), and COVID-19 VoC period (other vs delta vs omicron). For Cohort B (booster vaccine), we also investigate the effect of a previous mRNA vaccine scheme (yes vs no) as a subgroup. Subgroup analyses were evaluated with an interaction term between the vaccine and the subgroup of interest.

Sensitivity analyses were performed and included: stricter cancer definition, only tested patients (any test during the whole period), only RT-PCR COVID-19 diagnosis, and modified COVID-19 hospitalization outcome to any COVID-19 diagnosis from 21 days before admission to 3 days after, and from 14 days before admission to 3 days after, to exclude potential hospital-acquired COVID-19 infections and non-COVID-19 directly related hospitalizations respectively. Finally, to better address health bias, we built two additional matched cohorts for both exposures (primary vaccination and booster vaccination), including previous influenza vaccine receipt as exact matching, excluding patients hospitalized a month prior to the matching date, and propensity matching on a number of outpatient visits as numerical and not categorical variable (*restricted matching* cohorts). We compared COVID-19 and non-COVID-19 outcomes between matched cohorts (original versus restricted matching cohorts). The decision to evaluate different matching was made post-hoc during peer review.

We report 95% CI for all estimates. A p-value less than 5% was considered statistically significant. We performed all analyses in R version 3.6.0 (R Foundation for Statistical Computing, Vienna, Austria).

### Reporting summary
Further information on research design is available in the Nature Portfolio Reporting Summary linked to this article.

## Data availability
In accordance with the current European and national law, the data used in this study are only available for the researchers participating in this study. Thus, we are not allowed to distribute or make publicly available the data to other parties. However, researchers from public institutions can request data from SIDIAP if they comply with certain requirements. Further information is available online (https://www.sidiap.org/index.php/menu-solicitudesen/application-proccedure) or by contacting SIDIAP (sidiap@idiapjgol.org).

## Code availability
R scripts were made available to ensure the reproducibility of results and in accordance with good research practice (https://github.com/felippelazar/SIDIAP-CovidVaccineCancer/)[72].

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

## Acknowledgements

We thank all healthcare professionals in Catalonia who daily register information in the populations' electronic health records; the Institut Català de la Salut and the Programa d'analítica de dades per a la recerca i la innovació en salut for providing access to the different data sources accessible through SIDIAP. OTR acknowledges support from the END-VOC Project (Horizon 2021-2024), funded by the European Union under grant agreement no. 101046314. This manuscript is an honest, accurate, and transparent account of the study being reported. No important aspects of the study have been omitted. The funding source had no role during the design, analysis or writing of the current study.

## Author contributions

F.L.N., O.T.R., and T.D.S. were involved in the conceptualization, methodology, and data curation. F.L.N., O.T.R., G.C.J., N.M.B., B.R., and L.P.C. in formal analysis and investigation. F.L.N and O.T.R were involved in writing—the original draft. All authors were involved in writing—review, and editing.

## Competing interests

GCJ reports outside the scope of this paper to have received honorariums from AstraZeneca, Pfizer, Merck, Bristol-Myers, Novartis, Roche, Amgem, Janssen, Lilly, Takeda, Daiichi Sankyo, in addition to playing advisory roles to Boehringer, Pfizer, Bayer, Roche, Merck, Bristol-Meyers, AstraZeneca, Yuhan, Janssen, Libbs, Sanofi, Novartis, Lilly, Takeda, and Daiichi Sankyo. The remaining authors have no conflicts of interest to disclose.
