## [Peer Review File · Nature Communications]

Effectiveness of COVID-19 vaccines against severe COVID-19 among patients with cancer in Catalonia, SpainREVIEWER COMMENTS

Reviewer #1 (Remarks to the Author):

Overall, I feel this is a useful contribution and well executed. The authors use similar methodology for evaluating covid vaccine effectiveness as a number of prior papers and appear to execute this methodology well. I have some specific comments below.

It would be useful to show subgroup analyses by cancer treatment data, if available, since prior work shows this substantially affects vaccine effectiveness.

For cancer type, not sure why lung is the only one shown.

It would also be useful to show subgroup analysis by vaccine type since it is known that there are differences of effectiveness among them, so the VE estimates will be more useful if stratified by vaccine type.

Methods being moved to end requires reworking the results so the paper can be read linearly to at least some extent. E.g., need to explain what cohort A and B are.

The definition of severe covid as only based on the presence of a positive test in a window around a hospitalization/death is limited and now a bit out of date, as it is recognized that frequently hospitalizations/deaths with covid are not the same as hospitalization/deaths from covid. More precise definitions are for example discussed in this paper, among others:

<https://pubmed.ncbi.nlm.nih.gov/35012694/> I feel using some such definition would strengthen confidence in the results quite a bit. With the current loose definition, the authors are likely underestimating the VE against severe disease, since effectiveness against truly severe disease but as much not against infection, combined with the prior condition of cancer (which will often require hospitalization), implies there would be expected to be a larger percentage of "with covid but non-severe" hospitalizations among the vaccinated cancer patients than among the unvaccinated cancer patients.

I don't entirely understand the rationale for this: "Patients who had COVID-19 infection, but did not hospitalize nor die, were censored 28 days after the date of infection to account for a window of susceptibility to events of interest".

Also if you are viewing covid-19 infection without severe disease as a competing risk, shouldn't competing risk methods be used instead of simple censoring? Death is also a competing risk and is particularly important in cancer patients.

There is a typo in suppl fig 1 and 2 (missing digit in patient count in first box). Also are "dead" and "moved out" relative to what date? The text doesn't mention the death criterion that I see, and it mentions "moved out" as "Patients that were transferred out of SIDIAP before matching". But if the date "moved out" is evaluated is the match date, how can this exclusion criterion be applied so high up in the flow chart, before the matching takes place (esp for unvaccinated, who don't have a match date until they are matched).

Reviewer #2 (Remarks to the Author):

Yes, I do think your results are noteworthy and it is helpful. It is significant to the field and there is originality.

I do think that it would be worthy of publication if a few minor amendments were made:-

1) I'd be grateful if you could discuss your findings in reference to this large UK dataset, and consider referencing, as it is the biggest cohort to date- <https://www.nature.com/articles/s41598-023-36990-9>

2) I think you may benefit in potentially being a bit more cautious about the findings of no

"waning", as it is possible that the protection isn't waning, because there is also more infections happening, which may increase immunity. Also there may well be subgroups, like blood cancers where waning is observed.

3) In the conclusion please comment on if measures like long acting prophylaxis was used in Spain, like evusheld and if this should be updated and continued.

4) I'd be grateful if you could note that vaccine effectiveness is also partly contributed to behaviour. If the cancer population were to immediately stop all shielding and move to unrestrained contact/gatherings, this may well affect your results.

Reviewer #3 (Remarks to the Author):

Thank you for letting me review this interesting observational study which, among 1:1 matched pairs of vaccinated vs. non-vaccinated (N=92372 pairs (cohort A))/non-boosted (N=54267 pairs (cohort B)) adult Catalanian cancer patients, in an emulated target trial analysis estimated the effectiveness of COVID-19 vaccines against a composite outcome of COVID-19 hospitalization or COVID-19 death. The manuscript is clear, well written and well organized. In the review below I present major concerns, minor concerns and finally the specific questions from the journal are addressed.

Major concerns

The inherent and important high risk of healthy-wealthy-vaccinée bias when evaluating vaccine effects in observational studies has not been introduced, or discussed in the manuscript, nor thoroughly addressed in the analyses. Although the authors carried out matching (where we can see that matched cancer patients seem to have less metastatic disease (healthy vaccinée bias) and more control visits (better health seeking behavior)), provided relevant information on the timing of the observed vaccine-effect, and provided negative control outcomes, the risk of residual confounding remain high and must be better addressed. Due to obvious healthy and wealthy vaccinée bias observed in prior COVID-19 vaccine studies we know for example that inclusion of mortality not related to Covid-19 in all observational Covid-19 vaccine studies is essential to provide context.

Further, the negative control outcomes chosen for the present study do not really represent relevant negative control outcomes. Relevant negative control outcomes should signify general health (or wealth) differences, and differences in health seeking behavior between the comparison groups, for example:

- Mortality not related to COVID-19.
- All-cause admission.
- All-cause out-patient visits (Suppl. Table 2 shows clear difference between matched and unmatched).
- Use of antibiotics or other frequently used medication.
- Visits at the family physician.

Another major concern is the choice of primary outcome:

- Why did the authors decide on a primary composite outcome of COVID-19 hospitalization or death? Two co-primary outcomes of COVID-19 death, and hospitalization (please see below) must be provided separately, and only separately.
- Was the study protocol published or can the authors provide other documentation regarding preplanned outcomes and statistical analyses plan?
- I acknowledge that in observational studies COVID-19 as a cause of death may be difficult to define, so death within 28 days after a COVID-19 positive test may be an acceptable outcome measure – but were all accidents, traumas, and suicides excluded?
- But why did the authors decide on the rather unspecific outcome definition of COVID-19 hospitalization as any hospitalization within 28 days after a COVID-19 test? Definitely, this is too un-specific. Hospitalization within 14 days after a positive COVID-19 test may be considered acceptable, in combination with estimating the vaccine effect on a more specific COVID-19 hospitalization outcome based on the COVID-19-specific ICD10 codes B342A and B972A too.
- The clear limitations in the outcome definitions must be discussed in the manuscript.

The shifting VOC pose a separate challenge. For example, the relevance of the estimation in the present study of protection by first and second vaccine shots already a bit historical, however, the data may also be interpreted as reflecting a relevant and important immunocompetence (i.e., vaccine "take") in the population of cancer patients.

Minor concerns

I suggest adding the word COVID-19 to the title just before hospitalization.

Line 99: Please explain why nursing home residents were excluded.

Regarding baseline characteristics the description lines 111-116 suggest healthy and wealthy vaccinee bias. Although not surprising, such signal is serious and must be elaborated upon. The authors cannot take for granted that matching eliminates the important differences between the comparison groups.

Line 123: Provide proportion instead of the word small.

Figure 1: Better placed in the Supplement.

Figure 3: How do the authors explain the negative effect observed as increased risk of death among partially vaccinated individuals? How do the authors explain the sudden shift towards a negative vaccine effect after 119 days?

eMethods line 13: Dispensed medications in primary care may serve as a control outcome.

eMethods line 34: If information is available regarding active anti-cancer treatment leading to potential immune suppression and decline in vaccine immunogenicity, such information should be included.

Specific journal questions

- What are the noteworthy results?

Answer: It is important to estimate vaccine efficiency in populations which are typically not included in the efficacy trials. The present study did that, but the methodology needs to be stricter to secure reliable results with as little as possible healthy and wealthy vaccinee bias.

- Will the work be of significance to the field and related fields? How does it compare to the established literature? If the work is not original, please provide relevant references.

Answer: If the authors comply with the suggestions of the present review, the work will provide important information to the field.

- Does the work support the conclusions and claims, or is additional evidence needed?

Answer: Additional evidence is needed, as also specified above.

- Are there any flaws in the data analysis, interpretation and conclusions? Do these prohibit publication or require revision?

Answer: The topic of healthy-wealthy vaccinee bias must be elaborated upon in text and analyses. The primary outcome must not be composite. The co-primary outcome of hospitalization should be re-defined as described above, and a more specific COVID-19 hospitalization outcome must be added (ICD10 codes B342A and B972A)

- Is the methodology sound? Does the work meet the expected standards in your field?

Answer: The methodology is sound, except from the concerns mentioned in the present review.

- Is there enough detail provided in the methods for the work to be reproduced?

Answer: Yes.

REVIEWER COMMENTS

Reviewer #1 (Remarks to the Author):

Overall, I feel this is a useful contribution and well executed. The authors use similar methodology for evaluating covid vaccine effectiveness as a number of prior papers and appear to execute this methodology well. I have some specific comments below.

ANSWER: Thank you for reviewing our work and for the endorsement of our methods.

It would be useful to show subgroup analyses by cancer treatment data, if available, since prior work shows this substantially affects vaccine effectiveness.

ANSWER: Thank you for the commentary. Unfortunately, we do not have treatment data available. We tried to overcome this limitation by providing vaccine effectiveness by time from cancer diagnosis which may provide some insights as patients with recent diagnosis are more likely to be under active treatments (either curative or palliative treatments).

For cancer type, not sure why lung is the only one shown.

ANSWER: In addition to lung cancer, we provided sub-group analysis for patients with hematological cancers as well. We defined the subgroups to be tested prior to data analysis. We chose patients with lung cancer because of the higher risk of death following COVID-19 infection (1, 2); and patients with hematological cancer because of the previous evidence of lower vaccine effectiveness against COVID-19 infection reported (3, 4). Additionally, there is the risk of small sample sizes and lack of power. Because of the exploratory nature of sub-group analysis and the higher likelihood of false positive findings when multiple testing, we restricted our analysis to these two tumors, which we consider to be of greatest interest.

References:

1. Khoury, Emma, Sarah Nevitt, William Rohde Madsen, Lance Turtle, Gerry Davies, and Carlo Palmieri. "Differences in Outcomes and Factors Associated With Mortality Among Patients With SARS-CoV-2 Infection and Cancer Compared With Those Without Cancer: A Systematic Review and Meta-Analysis." *JAMA Netw Open* 5, no. 5 (May 2, 2022): e2210880. <https://doi.org/10.1001/jamanetworkopen.2022.10880>.
2. Maxwell Salvatore et al., "COVID-19 Outcomes by Cancer Status, Site, Treatment, and Vaccination," *Cancer Epidemiol. Biomarkers Prev.* 32, no. 6 (June 1, 2023): 748–59, <https://doi.org/10.1158/1055-9965.EPI-22-0607>.
3. Lennard Y W Lee et al., "COVID-19: Third Dose Booster Vaccine Effectiveness against Breakthrough Coronavirus Infection, Hospitalisations and Death in Patients with Cancer: A Population-Based Study," *Eur. J. Cancer* 175 (November 2022): 1–10, <https://doi.org/10.1016/j.ejca.2022.06.038>.
4. Lennard Y W Lee et al., "Vaccine Effectiveness against COVID-19 Breakthrough Infections in Patients with Cancer (UKCCEP): A Population-Based Test-Negative Case-

Control Study,” *Lancet Oncol.* 23, no. 6 (June 2022): 748–57, [https://doi.org/10.1016/S1470-2045\(22\)00202-9](https://doi.org/10.1016/S1470-2045(22)00202-9).

It would also be useful to show subgroup analysis by vaccine type since it is known that there are differences of effectiveness among them, so the VE estimates will be more useful if stratified by vaccine type.

ANSWER: Thank you for pointing this out. We decided to not present VE estimates by vaccine subtype in the primary series vaccination for two reasons. First, the vast majority of patients received mRNA vaccines (76%), leaving only a smaller proportion to other products. Second, vaccine product types in Spain were administered according to product availability and targeted risk groups; therefore, substantial population differences according to the vaccine product type were observed in terms of demographic and calendar time, which may influence these results. For example, in *Suppl. Figure 2*, we show that the majority of ChAdOx1 recipients were patients aged in-between 60 and 69 years old; consequently, any sub-group analysis of ChAdOx1 effectiveness would be confounded by the age of participants in this group, not allowing useful comparison with other vaccine types, since there will be no contrast. For this reason, we opted not to do sub-group analysis by vaccine type. For the booster dose, we provided a sub-group analysis based on previous vaccine type (mRNA vs non-mRNA). We agree this is a limitation, and we highlighted this limitation in the discussion section.

“Because of the restricted range of ages (mostly ≤ 69 years) that received initial ChAdOX1 vaccination, results might be not generalizable to all adults with cancer, particularly the older ones that did not receive ChAdOX1.” (Discussion Section, p. 8)

Methods being moved to end requires reworking the results so the paper can be read linearly to at least some extent. E.g., need to explain what cohort A and B are.

ANSWER: Thank you for the commentary. As suggested by the reviewer, we provided additional context information for both cohorts in the results section.

‘We built two matched cohorts: 184,744 patients (92,372 matched pairs) were included in the first and second dose (primary) vaccination cohort (Cohort A, Suppl. Figure 3) and 108,534 (54,267 matched pairs) in the booster vaccination cohort (Cohort B, Suppl. Figure 4)’ (Results Section, p. 4)

The definition of severe covid as only based on the presence of a positive test in a window around a hospitalization/death is limited and now a bit out of date, as it is recognized that frequently hospitalizations/deaths with covid are not the same as hospitalization/deaths from covid. More precise definitions are for example discussed in this paper, among others: <https://pubmed.ncbi.nlm.nih.gov/35012694/> I feel using some such definition would strengthen confidence in the results quite a bit. With the current loose definition, the authors are likely underestimating the VE against severe disease, since effectiveness against truly severe

disease but as much not against infection, combined with the prior condition of cancer (which will often require hospitalization), implies there would be expected to be a larger percentage of “with covid but non-severe” hospitalizations among the vaccinated cancer patients than among the unvaccinated cancer patients.

ANSWER: The reviewer is correct and we discussed this topic during our protocol planning. Unfortunately, we have no access to the main reason for hospitalization or the cause of death to provide more specific context into the hospitalizations and death outcomes, therefore we operationalize the definition as per other studies, including clinical trials and vaccine effectiveness (1, 2). To further improve this potential misclassification, we added a new outcome called ‘severe hospitalization’ which included patients hospitalized with COVID-19 and the need of any invasive oxygen supplementation (mechanical ventilation or tracheostomy or ECMO - see *Figure 2 and Figure 3 in the main article*). As pointed out by the reviewer, although the number of events are low, the results suggest a higher effectiveness when compared to only hospitalizations.

'Likewise, death was classified as any cause of death occurring within 28 days from the date of COVID-19 diagnosis. Secondary outcomes include COVID-19 hospitalization, COVID-19 severe hospitalization - defined as a COVID-19 hospitalization with the need of invasive oxygen supplementation, and COVID-19 death separately.' (Methods Section, p. 12)

1. Otavio T Ranzani et al., “Effectiveness of the CoronaVac Vaccine in Older Adults during a Gamma Variant Associated Epidemic of Covid-19 in Brazil: Test Negative Case-Control Study,” *BMJ*, August 20, 2021, n2015, <https://doi.org/10.1136/bmj.n2015>.
2. J. Daniel Kelly et al., “Comparative mRNA Booster Effectiveness against Death or Hospitalization with COVID-19 Pneumonia across at-Risk US Veteran Populations,” *Nature Communications* 14, no. 1 (May 23, 2023): 2976, <https://doi.org/10.1038/s41467-023-38503-8>.

I don’t entirely understand the rationale for this: “Patients who had COVID-19 infection, but did not hospitalize nor die, were censored 28 days after the date of infection to account for a window of susceptibility to events of interest”.

Also if you are viewing covid-19 infection without severe disease as a competing risk, shouldn’t competing risk methods be used instead of simple censoring? Death is also a competing risk and is particularly important in cancer patients.

ANSWER: Thank you for the commentary. The rationale for censoring COVID-19 infection is 1) we are evaluating first COVID-19 infection associated events, and 2) explained by the lower susceptibility to subsequent infections and consequently lower risk of the outcome after a window of susceptibility after the event. A patient is still at risk during COVID-19 infection (and subsequent window of susceptibility to the event), but this risk decreases if no event occurs and immunity is created.

Regarding competing risk analysis, the current analysis accounts for competing events for non-COVID-19 deaths when analyzing COVID-19 hospitalization, death and the combined outcome. Within the framework of competing risks, two main approaches exist: the Fine-Gray approach (subdistribution HR), more suitable for prediction and cause-specific approach (cause-specific HR), more suitable for aetiological/causation. We agree we can present both to better interpret the results. Because of the potential problems on interpreting Fine-Gray models when looking for associations (1-4), we stick to our original plan on using cause-specific HR, and present it in the supplement (*Suppl. Table 3*).

Nevertheless, interpretation remains the same. (*See table below*).

	VE (from HR) [cause-specific model]	VE (from subdistribution HR) [Fine- Gray model]
Primary Vaccination		
Partially Vaccinated	37.7% (17.6 - 52.8)	38.2% (28.5 - 46.6)
Fully Vaccinated	53.1% (42.3 - 61.8)	64.3% (56.8 - 70.5)
Booster Vaccination		
14 - 60 days after Booster	78.7% (71.0 - 84.3)	58.7% (50.4 - 65.7)
> 60 days after Booster	52.9% (39.0 - 63.6)	72.7% (67.6 - 77.1)

'In a post-hoc decision during the peer-review, we also estimated the VE using the Fine-Gray models considering the competing risk of death, deriving sub-distribution HRs for the main analysis, an approach more suitable for prediction and prognostic, to complement the cause-specific evaluation. The competing event was non-COVID-19 deaths for all models, except when the outcome was COVID-19 hospitalization, when the competing event was all-cause death' (Methods Section, p.13)

'Competing hazards model (non-COVID death as competing risk) showed comparable results (Suppl. Table 3)' (Results Section, p. 5)

References

1. Marcel Wolbers et al., "Competing Risks Analyses: Objectives and Approaches," *European Heart Journal* 35, no. 42 (November 7, 2014): 2936–41, <https://doi.org/10.1093/eurheartj/ehu131>.
2. Per Kragh Andersen et al., "Competing Risks in Epidemiology: Possibilities and Pitfalls," *International Journal of Epidemiology* 41, no. 3 (June 2012): 861–70, <https://doi.org/10.1093/ije/dyr213>.

3. Marcel Wolbers et al., “Prognostic Models With Competing Risks: Methods and Application to Coronary Risk Prediction,” *Epidemiology* 20, no. 4 (July 2009): 555–61, <https://doi.org/10.1097/EDE.0b013e3181a39056>.
4. Aurelien Latouche et al., “A Competing Risks Analysis Should Report Results on All Cause-Specific Hazards and Cumulative Incidence Functions,” *Journal of Clinical Epidemiology* 66, no. 6 (June 2013): 648–53, <https://doi.org/10.1016/j.jclinepi.2012.09.017>.

There is a typo in suppl fig 1 and 2 (missing digit in patient count in first box). Also are “dead” and “moved out” relative to what date? The text doesn’t mention the death criterion that I see, and it mentions “moved out” as “Patients that were transferred out of SIDIAP before matching”. But if the date “moved out” is evaluated is the match date, how can this exclusion criterion be applied so high up in the flow chart, before the matching takes place (esp for unvaccinated, who don’t have a match date until they are matched).

ANSWER: We thank the reviewer for pointing this out. We added the missing number in the typo pointed out by the reviewer and clarified the other appointments. The cohort construction was dynamic, emulating a clinical trial on a daily basis (i.e. dynamically constructing those eligible and potentially able to be “randomized”). The initial population included all patients with a cancer diagnosis at least five years from the beginning of vaccination date (27/Dec/2015). An initial filter was applied to those that had moved-out or died before the beginning of the vaccination campaign (27/Dec/2020) as they would not be eligible for vaccination (Suppl. Figure 3 and Suppl. Figure 4, box 4). During vaccine rollout, we applied another filter in patients that have moved-out or died on a daily basis. For example, a patient could be alive at the beginning of the vaccination campaign, but could have died in the following days after vaccination start, before being vaccinated or before being matched as control; and therefore excluded during REM (box 7). We better clarified this issue in both figures with a legend.

Reviewer #2 (Remarks to the Author):

Yes, I do think your results are noteworthy and it is helpful. It is significant to the field and there is originality.

ANSWER: Thank you very much for the time spent and insights provided to our manuscript.

I do think that it would be worthy of publication if a few minor amendments were made:-

- 1) I'd be grateful if you could discuss your findings in reference to this large UK dataset, and consider referencing, as it is the biggest cohort to date- <https://www.nature.com/articles/s41598-023-36990-9>

ANSWER: Thank you for the suggestion. The referenced article is interesting, and provides appealing evidence about the reduction in case-fatality COVID-19 cases after vaccination in cancer patients. We added its reference to the introduction.

'Population UK data indicates a reduced temporal cancer COVID-19 fatality rate after vaccination, though still higher than in healthy individuals' (Introduction Section, p. 3)

2) I think you may benefit in potentially being a bit more cautious about the findings of no "waning", as it is possible that the protection isn't waning, because there is also more infections happening, which may increase immunity. Also there may well be subgroups, like blood cancers where waning is observed.

ANSWER: Thank you for the commentary. We better clarified the limitations of waning analysis in light of the reviewer comment.

Although results are promising, they should be interpreted having in mind the potential bias introduced by susceptibles depletion⁴³ and undocumented infections.' (Discussion Section, p.8)

3) In the conclusion please comment on if measures like long acting prophylaxis was used in Spain, like evusheld and if this should be updated and continued.

ANSWER: Thank you. We are unaware of any national campaign of mechanical prophylaxis other than masks during the pandemic. We agree with the reviewer that patients at high-risk may have used alternative mechanical protections such as 'face-shields', and add a sentence in the limitation section highlighting this.

'Additionally, during the COVID-19 pandemic, patients at higher risk of serious outcomes might have taken additional measures to prevent infection such as avoiding gatherings or using face-shields which are not captured by data and may influence results' (Discussion Section, p. 10)

4) I'd be grateful if you could note that vaccine effectiveness is also partly contributed to behaviour. If the cancer population were to immediately stop all shielding and move to unrestrained contact/gatherings, this may well affect your results.

ANSWER: Thank you for pointing this out. As suggested, we added a sentence in the limitations paragraph highlighting this issue.

'Additionally, during the COVID-19 pandemic, patients at higher risk of serious outcomes might have taken additional measures to prevent infection such as avoiding gatherings or using face-shields which are not captured by data and may influence results' (Discussion Section, p. 10)

Reviewer #3 (Remarks to the Author):

Thank you for letting me review this interesting observational study which, among 1:1 matched pairs of vaccinated vs. non-vaccinated (N=92372 pairs (cohort A))/non-boosted (N=54267 pairs (cohort B)) adult Catalonian cancer patients, in an emulated target trial analysis estimated the effectiveness of COVID-19 vaccines against a composite outcome of COVID-19 hospitalization

or COVID-19 death. The manuscript is clear, well written and well organized. In the review below I present major concerns, minor concerns and finally the specific questions from the journal are addressed.

ANSWER: Thank you for reviewing our paper. We appreciate the time and effort spent.

Major concerns

The inherent and important high risk of healthy-wealthy-vaccinée bias when evaluating vaccine effects in observational studies has not been introduced, or discussed in the manuscript, nor thoroughly addressed in the analyses. Although the authors carried out matching (where we can see that matched cancer patients seem to have less metastatic disease (healthy vaccinée bias) and more control visits (better health seeking behavior)), provided relevant information on the timing of the observed vaccine-effect, and provided negative control outcomes, the risk of residual confounding remain high and must be better addressed. Due to obvious healthy and wealthy vaccinée bias observed in prior COVID-19 vaccine studies we know for example that inclusion of mortality not related to Covid-19 in all observational Covid-19 vaccine studies is essential to provide context.

Further, the negative control outcomes chosen for the present study do not really represent relevant negative control outcomes. Relevant negative control outcomes should signify general health (or wealth) differences, and differences in health seeking behavior between the comparison groups, for example:

- Mortality not related to COVID-19.
- All-cause admission.
- All-cause out-patient visits (Suppl. Table 2 shows clear difference between matched and unmatched).
- Use of antibiotics or other frequently used medication.
- Visits at the family physician.

ANSWER: Thank you for highlighting this issue. We agree with the reviewer that this topic could be expanded in our analysis and better discussed.

As suggested by the reviewer, we calculated the vaccine effectiveness for all-cause hospitalization and non-COVID death for both cohorts (*Suppl. Table 5 and 6, Table AR1*). We observed only a small protection against all-cause hospitalization right after the matching and larger protection against non-COVID deaths. For non-COVID deaths, we observed sustained risk reduction after vaccination for both cohorts, but with larger treatment effects for Cohort B.

This phenomenon has been known in vaccine effectiveness studies for a while, even before COVID-19 (such as for Influenza). Several mechanisms might be involved, including 1) an indirect effect of the vaccine on other causes (phenomenon observed even in RCTs of COVID-19 vaccines (1); 2) deaths from COVID-19 not diagnosed among the non-COVID-19 deaths, which might be amplified in cancer patients in palliative care not being tested; 3) healthy-vaccine bias, therefore we added in the pool of eligible to match individuals that would not be

eligible to be randomized in a RCT due to extreme frailty, a known problem in influenza studies (2) or severe immunosuppression, or hospitalized, among others.

References:

1. Christine S. Benn et al., “Randomized Clinical Trials of COVID-19 Vaccines: Do Adenovirus-Vector Vaccines Have Beneficial Non-Specific Effects?,” *iScience* 26, no. 5 (May 2023): 106733, <https://doi.org/10.1016/j.isci.2023.106733>.
2. Lisa A Jackson et al., “Functional Status Is a Confounder of the Association of Influenza Vaccine and Risk of All Cause Mortality in Seniors,” *International Journal of Epidemiology* 35, no. 2 (April 1, 2006): 345–52, <https://doi.org/10.1093/ije/dyi275>.

To better explore these findings, we built new matching cohorts 1) using a more restricted matching criteria considering the previous receipt of influenza vaccine as exact matching, to better capture additional healthy-vaccinee bias not captured by previous health-system visits and 2) excluding patients hospitalized a month prior to vaccine receipt / eligibility. We calculated vaccine protection against all-cause hospitalization and non-COVID deaths in the same time-periods as the original cohorts (see *Table AR2 below*). In this analysis, the healthy-vaccinee bias was better controlled for all-cause hospitalization, virtually not experiencing this bias (*Suppl. Figures 10 and 11*). Nevertheless, the protection against non-COVID-19 deaths remained high. Interestingly, in the original cohorts, we observed a higher proportion of patients experiencing non-COVID-19 deaths without any preceding hospitalization when compared to COVID-19 deaths (see *Table AR3*), suggesting a possibility of undocumented COVID-19 deaths in this group, although not all explained by this. Thus, we could explain part of this bias by restricting to a potentially less sick cohort and further controlling for healthy-vaccinee bias. A full explanation of the phenomenon of protection against non-COVID-deaths remains uncertain as discussed in other papers and we believe we did further advancements on aiming to understand it. In an oncological cohort, where the baseline risk of dying by any cause is much higher than the general population, this bias can be seen as amplified, but not directly translating to biasing vaccine effectiveness. Indeed, the oncological patient that was not vaccinated might have more behavior protection against infection, might be so frail that also was not tested by COVID-19 (Cohort A 70% and Cohort B 61% of non-COVID deaths occurred without hospitalization).

Further exploring this topic, and as suggested by the reviewer, we calculated healthcare usage outcomes (See *Suppl. Table 8 and Table AR4 below*). We observed a decreased utilization of most healthcare services in the immediate period post-vaccination for the vaccine group; however, after the immediate period post-vaccination, results were more mixed: we found a slight increase in non-serious health services usage for the vaccine group (outpatients visits, tele-health) and a decrease (protection) for more serious outcomes (inpatient, UCI admission). However, these findings were heterogeneous between primary and booster cohorts, suggesting a different matched-control population between them or different calendar times pandemic scenarios, since during the first/second doses, part of the health system was not properly working. These findings may still suggest some confounding associated with health-seeking behavior and healthy bias respectively, which we now address in the discussion section.

Finally, as suggested by the reviewer, we calibrated our estimates with a more expanded number of negative outcomes to include medication prescriptions and other possibly unrelated health issues (low back pain, mild depression, urinalysis, mammography, chest pain, irregular periods, migraine, acetaminophen, amoxicillin and ibuprofen, see *Table AR5 below*). These additional negative outcomes provided very similar calibrated results to what we observed in the original analysis and, therefore, we decided to keep our original analysis because of the previous scientific validation of these negative outcomes. We updated our methods, results and discussion accordingly.

'Vaccination was associated with decreased hazards of all-cause hospitalizations and non-COVID deaths in the immediate period post-vaccination for both cohorts (Suppl. Tables 5 and 6). For Cohort A, primary vaccination was associated with a non-significant decrease in all-cause hospitalizations during follow-up but a sustained decreased hazard for non-COVID death (Suppl. Table 5). For Cohort B, booster vaccination was associated with lower risk of all-cause hospitalizations and non-COVID death in all time periods (Suppl. Table 6). We observed a lower proportion of non-COVID-19 deaths without preceding hospitalization than COVID-19 deaths (Suppl. Table 7). Analysis of health services usage (outpatient, telehealth, home, inpatient and ICU visits) by vaccination status showed lower likelihood in the vaccination group for most outcomes in cohort B, but not for Cohort A; which showed increased hazards for tele-health, home and outpatient visits and lower hazards for inpatient and ICU visits (Suppl. Table 8).' (Results Section, p. 6)

'Sensitivity analysis including previous influenza vaccine receipt in matching and excluding those hospitalized a month prior to vaccination in the primary and booster cohorts (restricted matching cohort, Suppl. Tables 9 and 10) reduced the vaccine protection for all-cause hospitalizations, but not non-COVID death, which remained lower in the vaccinated group (Suppl. Tables 11 and 12, Suppl. Figures 10 and 11). The COVID-19 primary outcome had comparable results (Suppl. Table 13)' (Results Section, p. 6)

'Estimation of VE from observational studies⁴⁴ is challenging as vaccinated patients are often healthier (healthy bias) and more health conscious (health-seeking bias) than unvaccinated individuals. Although proper matching on relevant covariates may provide well balanced characteristics between groups, it is often insufficient to adjust for unmeasured confounders. Previous research on COVID-19 vaccine effectiveness has shown that, despite proper matching, vaccinated patients had lower risk of non-COVID death compared to control^{44–46} partially explained by healthier conditions of vaccinated patients. In our study, we showed a lower risk of non-COVID deaths among vaccinated patients but much lower magnitude for all-cause hospitalizations. Further adjustment for previous influenza receipt and exclusion of recently hospitalized patients, variables that would capture part of the healthy and health-seeking biases, improved all cause-hospitalization differences between groups, but not non-COVID death. We hypothesize three other reasons for this finding, beyond bias. First, the principal difference between vaccinated and control ones occurred in the immediate period after the vaccine (Suppl. Table 5 and 6), as observed by other studies. This is likely because the most ill or frail individuals ended-up dying early on after matching, and in a clinical trial they

would likely not be eligible to be randomized. Second, part of the non-COVID-19 deaths could be actually COVID-19 deaths, leading to misclassification of the outcome. In the analyzed population, there is a high proportion of non-COVID-19 deaths that occurred without hospitalization (60% for Cohort A and 71% for Cohort B), which associated with a great network of home and palliative care in the region, makes us hypothesize that these individuals are unlikely to get tested (and consequently being diagnosed). Third, indirect effects of COVID-19 vaccines on non-COVID-19 outcomes, still an open field of research. The protection against non-COVID-19 deaths has been observed even when analyzing clinical trials of adenovirus COVID-19 vaccines⁴⁷. Finally, socio-economic variables in large databases can not capture all socio-economic nuances and wealthy bias is a concern. The comparable risk in all-cause hospitalizations and similar effectiveness after negative outcomes calibration and across the sensitivity analyses, results in confidence in the vaccine protection observed, however the magnitude and direction of the potential bias is uncertain.' (Discussion Section, pp 8-9)

Table AR1: All Cause-Hospitalization (All-Hosp) and Non-COVID Death Outcomes (Original Analysis)

Outcome	Primary Vaccination (Cohort A) Periods	HR (95% Confidence Interval)	Booster Vaccination (Cohort B) Periods	HR (95% Confidence Interval)
All-Cause Hospitalization	no-vax	Ref	no-vax	Ref.
	V1 0-14D	0.68 (0.63 - 0.74)	V3 0-14D	0.64 (0.55 - 0.74)
	V1 14-59D	0.90 (0.82 - 0.98)	V3 14-28D	0.76 (0.63 - 0.92)
	V1 60D+	0.85 (0.72 - 1.01)	V3 28-60D	0.61 (0.52 - 0.73)
	V1V2 0-13D	0.89 (0.80 - 1.00)	V3 60-120	0.82 (0.71 - 0.95)
	V1V2 14-59D	1.01 (0.93 - 1.09)	V3 120+	0.93 (0.78 - 1.11)
	V1V2 60-89D	0.95 (0.83 - 1.08)	-	-
	V1V2 90-120D	1.14 (0.99 - 1.31)	-	-
	V1V2 120D+	1.47 (1.30 - 1.66)	-	-
Non-COVID Deaths	no-vax	Ref.	no-vax	Ref.
	V1 0-14D	0.07 (0.05 - 0.10)	V3 0-14D	0.04 (0.02 - 0.09)
	V1 14-59D	0.46 (0.39 - 0.55)	V3 14-28D	0.10 (0.05 - 0.18)
	V1 60D+	0.40 (0.30 - 0.53)	V3 28-60D	0.16 (0.11 - 0.22)
	V1V2 0-13D	0.05 (0.03 - 0.09)	V3 60-120	0.21 (0.16 - 0.28)
	V1V2 14-59D	0.29 (0.24 - 0.34)	V3 120+	0.33 (0.26 - 0.42)
	V1V2 60-89D	0.33 (0.26 - 0.42)		
	V1V2 90-120D	0.30 (0.24 - 0.38)		
	V1V2 120D+	0.44 (0.37 - 0.51)		

Table AR2: All Cause-Hospitalization (All-Hosp) and Non-COVID Death Outcomes (More Restricted Matching Cohort)

Outcome	Primary Vaccination (Cohort A) Periods	HR (95% Confidence Interval)	Booster Vaccination (Cohort B) Periods	HR (95% Confidence Interval)
All-Cause Hospitalization	no-vax	Ref.	no-vax	Ref.
	V1 0-14D	0.75 (0.68 - 0.83)	V3 0-14D	0.71 (0.60 - 0.85)
	V1 14-59D	0.94 (0.85 - 1.04)	V3 14-28D	0.75 (0.61 - 0.93)
	V1 60D+	1.14 (0.98 - 1.32)	V3 28-60D	0.77 (0.63 - 0.95)
	V1V2 0-13D	0.98 (0.87 - 1.12)	V3 60-120	0.87 (0.73 - 1.03)
	V1V2 14-59D	0.92 (0.84 - 1.01)	V3 120+	0.95 (0.77 - 1.17)
	V1V2 60-89D	0.83 (0.73 - 0.95)	-	-
	V1V2 90-120D	0.73 (0.64 - 0.83)	-	-
	V1V2 120D+	1.04 (0.95 - 1.13)	-	-
Non-COVID Deaths	no-vax	Ref.	no-vax	Ref.
	V1 0-14D	0.08 (0.05 - 0.12)	V3 0-14D	0.05 (0.02 - 0.14)
	V1 14-59D	0.52 (0.42 - 0.64)	V3 14-28D	0.10 (0.04 - 0.23)
	V1 60D+	0.35 (0.24 - 0.50)	V3 28-60D	0.09 (0.05 - 0.15)
	V1V2 0-13D	0.07 (0.04 - 0.14)	V3 60-120	0.18 (0.13 - 0.27)
	V1V2 14-59D	0.24 (0.19 - 0.31)	V3 120+	0.33 (0.24 - 0.45)
	V1V2 60-89D	0.35 (0.26 - 0.46)	-	-
	V1V2 90-120D	0.24 (0.18 - 0.32)	-	-
	V1V2 120D+	0.37 (0.31 - 0.45)	-	-

Table AR3: Types of Deaths and Proportion of Preceding Hospitalizations

Cohort	Any Hospitalization before non-COVID death	Any Hospitalization before COVID-19 death
Primary Vaccination Cohort	991/3292 (30%)	63/93 (67%)
Booster Cohort	447/1127 (39%)	71/134 (52%)

Table AR4: HealthCare Utilization by Periods After Vaccination (Original Cohorts)

Outcome	Primary Vaccination (Cohort A) Periods	HR (95% Confidence Interval)	Booster Vaccination (Cohort B) Periods	HR (95% Confidence Interval)
All-cause Outpatient Visit	Unvaccinated	Ref	Un-boosted	Ref
	0 - 14 days	0.45 (0.44 - 0.46)	0 - 14 days	0.42 (0.41 - 0.43)
	Partially Vaccinated	1.58 (1.55 - 1.61)	14 - 60 days after booster	0.46 (0.44 - 0.47)
	Fully Vaccinated	0.97 (0.95 - 1.01)	60 days+ after booster	1.00 (0.95 - 1.05)
All-cause Telehealth Visit	Unvaccinated	Ref	Un-boosted	Ref
	0 - 14 days	0.87 (0.86 - 0.89)	0 - 14 days	0.87 (0.84 - 0.90)
	Partially Vaccinated	1.07 (1.04 - 1.09)	14 - 60 days after booster	0.87 (0.84 - 0.90)
	Fully Vaccinated	1.16 (1.13 - 1.19)	60 days+ after booster	1.09 (1.03 - 1.15)
All-cause Home Visit	Unvaccinated	Ref	Un-boosted	Ref
	0 - 14 days	0.70 (0.66 - 0.75)	0 - 14 days	0.50 (0.46 - 0.54)
	Partially Vaccinated	1.37 (1.29 - 1.46)	14 - 60 days after booster	0.53 (0.48 - 0.59)
	Fully Vaccinated	0.95 (0.89 - 1.01)	60 days+ after booster	0.85 (0.76 - 0.95)
All-cause UCI Visit	Unvaccinated	Ref	Un-boosted	Ref
	0 - 14 days	0.57 (0.39 - 0.84)	0 - 14 days	0.50 (0.25 - 1.00)
	Partially Vaccinated	0.66 (0.50 - 0.87)	14 - 60 days after booster	0.36 (0.21 - 0.63)
	Fully Vaccinated	0.79 (0.64 - 0.96)	60 days+ after booster	1.26 (0.84 - 1.87)
All-cause Inpatient	Unvaccinated	Ref	Un-boosted	Ref
	0 - 14 days	0.74 (0.68 - 0.80)	0 - 14 days	0.69 (0.60 - 0.80)
	Partially Vaccinated	0.87 (0.81 - 0.94)	14 - 60 days after booster	0.67 (0.59 - 0.76)
	Fully Vaccinated	1.07 (1.01 - 1.14)	60 days+ after booster	0.86 (0.77 - 0.96)

Table AR5: Original results and Calibrated Outcomes with Two Different Sets of Negative Outcomes (Original NCO Outcomes N = 43 and Expanded NCO Outcomes)

	Original VE	Calibrated VE after Original NCO outcomes	Calibrated VE after Expanded NCO outcomes*
Primary Vaccination			
14 days after 1st Dose	37.7% (17.6 - 52.8)	38.3% (5.4 - 59.8)	39.7% (10.5 - 59.4)
7 days after 2nd dose	53.1% (42.3 - 61.8)	59.3% (45.6 - 69.6)	60.0% (44.4 - 71.1)
Booster Vaccination			
14 - 60 days after 3rd dose	78.7% (71.0 - 84.3)	74.4% (63.4 - 82.0)	73.0% (61.6 - 81.0)
60 or more days after 3rd dose	52.9% (39.0 - 63.6)	57.2% (12.5 - 79.0)	56.3% (15.4 - 77.4)

* = plus low back pain, mild depression, urinalysis, mammography, chest pain, irregular periods, migraine, acetaminophen, amoxicillin and ibuprofen

Another major concern is the choice of primary outcome:

- Why did the authors decide on a primary composite outcome of COVID-19 hospitalization or death? Two co-primary outcomes of COVID-19 death, and hospitalization (please see below) must be provided separately, and only separately.

ANSWER: We decided on a composite outcome due to the expected low number of events for each outcome separately (particularly for the outcome of COVID-19 death); and to deal with the previous knowledge that several cancer patients in Catalonia might not be hospitalized due to home-care/palliative care at home, therefore the composite outcome better dealing with this scenario. Although our primary outcome was a combined outcome, we provided separate estimates for each of the individual outcomes in the main article.

- Was the study protocol published or can the authors provide other documentation regarding preplanned outcomes and statistical analyses plan?

ANSWER: Thank you. Although the initial statistical analysis plan was pre-planned, we have not published it. We can assure the pre-planned analysis was followed and any deviation would be labeled as post-hoc.

- I acknowledge that in observational studies COVID-19 as a cause of death may be difficult to define, so death within 28 days after a COVID-19 positive test may be an acceptable outcome measure – but were all accidents, traumas, and suicides excluded?

ANSWER: Thank you for the commentary. We agree with the reviewer that information on the cause of death would provide substantial information, but, unfortunately, we did not have access to it (the cause of death is not available in the SIDIAP database). Additionally, we don't expect

these causes of death to be frequent in this population, not resulting in a misclassification problem.

- But why did the authors decide on the rather unspecific outcome definition of COVID-19 hospitalization as any hospitalization within 28 days after a COVID-19 test? Definitely, this is too un-specific. Hospitalization within 14 days after a positive COVID-19 test may be considered acceptable, in combination with estimating the vaccine effect on a more specific COVID-19 hospitalization outcome based on the COVID-19-specific ICD10 codes B342A and B972A too.

ANSWER: Thank you for pointing this out. We defined the COVID-19 diagnosis window for hospitalization from 21 days before admission to the date of discharge based on previous literature (1). Additionally, to define a COVID-19 event is not easy, and the time window can capture complications of the disease, such as a thromboembolic event, still preventable by the vaccine. We would like to note that, we performed sensitivity analysis with different COVID-19 hospitalization definitions including: (i) 21 days before admission until 3 days after admission; and (ii) including only laboratorial confirmed COVID-19. To better clarify this issue, as suggested by the reviewer, we performed a new analysis considering only patients with COVID-19 diagnosis from 14 days before admission until 3 days after admission, which can be found in Suppl. Figure 12 and 13. As shown, the different definitions resulted in very minor changes compared to the original definition.

References

1. Roel, E. et al. Characteristics and Outcomes of Over 300,000 Patients with COVID-19 and History of Cancer in the United States and Spain. *Cancer Epidemiol. Biomarkers Prev.* 30, 1884–1894 (2021).

- The clear limitations in the outcome definitions must be discussed in the manuscript. The shifting VOC pose a separate challenge. For example, the relevance of the estimation in the present study of protection by first and second vaccine shots already a bit historical, however, the data may also be interpreted as reflecting a relevant and important immunocompetence (i.e., vaccine “take”) in the population of cancer patients.

ANSWER: Thank you for the commentary. We better explored this issue in the limitations paragraph in the discussion section. We would like to highlight that it is missing information on VE for special populations, thus even if somehow historical, there is a need to better understand the dynamics and effectiveness of vaccination campaigns for special populations.

'We defined COVID-19 outcomes based on temporal association of a positive diagnosis and the outcome (hospitalization and death) as causes for hospitalization and death were unavailable; however, sensitivity analysis with different definitions provided similar results.' (Discussion Section, p. 9)

'Lastly, during the Omicron wave, we could not differentiate between patients who have been hospitalized with COVID-19 and not because of COVID-19, however this is even harder to

ascertain in oncological patients. We included a sensitivity analysis with COVID-19 diagnosis up to 3 days after admission with similar results, showing that time of COVID-19 infection (pre-admission vs during admission) did not change outcomes.' (Discussion Section, p. 10)

Minor concerns

I suggest adding the word COVID-19 to the title just before hospitalization.

ANSWER: Thank you. As suggested, we added COVID-19 before hospitalization in the title.

'Effectiveness of COVID-19 Vaccines Against COVID-19 Hospitalization and Death Among Cancer Patients in Catalonia, Spain: A Population-based Cohort' (Title, p. 1)

Line 99: Please explain why nursing home residents were excluded.

ANSWER: We opted to exclude nursing home residents from our cohort because of the different profile of such patients (usually more ill, fragile, and more prone to serious events compared to the general population) which could limit the generalizability of our findings. Additionally, the majority of them could not be eligible for hospitalization, thus creating a population not at risk for the event of interest. By excluding them, we aimed for a more homogeneous population of cancer patients, more likely to resemble the one found in routine clinical practice.

Regarding baseline characteristics the description lines 111-116 suggest healthy and wealthy vaccinee bias. Although not surprising, such signal is serious and must be elaborated upon. The authors cannot take for granted that matching eliminates the important differences between the comparison groups.

ANSWER: Thank you. We further explored this topic in a previous answer and made the necessary adjustments in the article as suggested by the reviewer.

'However, the decreased hazards of non-COVID death, particularly in the immediate period post-vaccination, possibly indicates residual healthy bias and results should be interpreted in light of these findings, as previously discussed.' (Discussion Section, p. 9)

Line 123: Provide proportion instead of the word small.

ANSWER: Thank you. We added the proportion as suggested by the reviewer.

'14% and 11% of patients had metastatic disease in Cohort A and B respectively.' (Results Section, p. 5)

Figure 1: Better placed in the Supplement.

ANSWER: Thank you. We added Figure 1 to Supplementary Material as Suppl. Figure 1.

Figure 3: How do the authors explain the negative effect observed as increased risk of death among partially vaccinated individuals? How do the authors explain the sudden shift towards a negative vaccine effect after 119 days?

ANSWER: Thank you for the commentary. We updated our estimates for COVID-19 death after we found a typo in the analysis code which resulted in minor changes in the number of events and results. The updated results did show a numerical negative effect in effectiveness after 120 days, but not significant. During follow-up, differential depletion of susceptibles (1) may introduce selection bias and affect VE estimates, possibly explaining the numerical decrease in effectiveness after 120 days.

References

1. Rebecca Kahn et al., "Identifying and Alleviating Bias Due to Differential Depletion of Susceptible People in Postmarketing Evaluations of COVID-19 Vaccines," *American Journal of Epidemiology* 191, no. 5 (March 24, 2022): 800–811

eMethods line 13: Dispensed medications in primary care may serve as a control outcome.

ANSWER: Thank you for the commentary. We explored this topic in a previous answer (*expanded negative control outcomes analysis, Table AR5*). The expanded control outcomes (including medications) provided very similar results therefore we decided to keep the original analysis.

eMethods line 34: If information is available regarding active anti-cancer treatment leading to potential immune suppression and decline in vaccine immunogenicity, such information should be included.

ANSWER: Thank you for the commentary. Unfortunately, we do not have this data available. However, we tried to overcome this limitation by providing vaccine effectiveness by time from cancer diagnosis which may provide some insights as patients with recent diagnosis are more likely to be under active treatments (either curative or palliative treatments).

REVIEWER COMMENTS

Reviewer #1 (Remarks to the Author):

The authors have addressed the concerns I raised and I don't have further comments.

Reviewer #3 (Remarks to the Author):

Thank you for letting me re-review this interesting and relevant observational study estimating the effectiveness of COVID-19 vaccines against a composite outcome of COVID-19 hospitalization or COVID-19 death. Below I repeat my major concerns, which have not been sufficiently addressed in the revised manuscript.

Major concerns

Although the high risk in observational vaccine efficiency studies of healthy-wealthy-vaccinée bias is now considered, the new results I requested which is now provided in the revision clearly demonstrate such healthy-wealthy-vaccinée bias in the present study, despite the authors' relevant attempts to diminish bias.

To exemplify this, please see Supplemental Figures 10 and 11.
Here it can be seen that:

1. Supplemental Figure 10.

Although COVID-19 vaccine is not associated with a decreased risk of all-cause hospitalization (A), the new results show COVID-19 vaccine to be associated with a clear and immediate protective effect against non-COVID-19 death (B), which cannot be true. Vaccine protection is based in an immunological reaction which takes at least two weeks. So, an immediate protective effect against non-COVID-death is not possible. The phenomenon can only be explained by bias, i.e., the vaccine recipients having an inherent better health and decreased risk of death compared with the unvaccinated population, even after matching. Assuming that vaccine effects are specific (since so-called "non-specific effects of vaccines" can be found in observational studies only (please see below)) against the target disease (here COVID-19 associated hospitalization and death) such finding of non-specific immediate beneficial "effects" clearly demonstrate that healthy-wealthy-vaccinée bias is present and that the matching procedure does not at all help. The comparison groups are not comparable when it comes to risk of death. For confirmation of my point, please see Supplemental Tables 11 and 12.

2. Supplemental Figure 11.

Here, COVID-19 vaccine booster is found to be associated with both protection against all-cause hospitalization (A), which is unlikely, and an immediate protection against non-COVID-19 death (B), which cannot be true. Again, restricted matching as suggested by the authors does not help at all. The comparison groups are not comparable when it comes to risk of hospitalization, and/or death, meaning that the "vaccine effect" is overestimated throughout the study. These observations demonstrate our inability to control for bias in observational studies and must be taken seriously. These observations have nothing to do with vaccine "effects" but are caused by inherent differences between the comparison groups. Of course, the COVID-19 vaccine decreases the risk of severe COVID-19; this was demonstrated in randomised controlled trials. However, it is clearly not easy to validly estimate the protective effect in an observational study like the present.

Please note that despite being persuaded intensively for more than 40 years, the hypothesis the authors refer to about vaccines potentially having "non-specific" effects cannot be confirmed by randomised trial data and is most likely based on – bias in observational studies.

3. The authors wish to stick to the composite outcome of COVID-19-hospitalisation and -death. I do not agree. First, the two outcomes are clearly different in frequency – please see Figure 2.

Second, as demonstrated by the supplemental Figures 10-11, the bias differs by outcome. So, these two outcomes cannot be lumped together. As suggested in my first review, the two outcomes must be presented separate only.

In summary, due to significant healthy-and-wealthy vaccinee bias the present study generally overestimate the COVID-19 vaccine protective effect. This well-known problem has not been properly addressed in the revised manuscript.

Reviewer #1 (Remarks to the Author):

The authors have addressed the concerns I raised and I don't have further comments.

ANSWER: Thank you.

Reviewer #3 (Remarks to the Author):

Thank you for letting me re-review this interesting and relevant observational study estimating the effectiveness of COVID-19 vaccines against a composite outcome of COVID-19 hospitalization or COVID-19 death. Below I repeat my major concerns, which have not been sufficiently addressed in the revised manuscript.

Major concerns

Although the high risk in observational vaccine efficiency studies of healthy-wealthy-vaccinée bias is now considered, the new results I requested which is now provided in the revision clearly demonstrate such healthy-wealthy-vaccinée bias in the present study, despite the authors' relevant attempts to diminish bias.

ANSWER: Thank you for revising our article one more time. We hope our answers are sufficient to suffix residual concerns the reviewer may still have.

To exemplify this, please see Supplemental Figures 10 and 11.

Here it can be seen that:

1. Supplemental Figure 10.

Although COVID-19 vaccine is not associated with a decreased risk of all-cause hospitalization (A), the new results show COVID-19 vaccine to be associated with a clear and immediate protective effect against non-COVID-19 death (B), which cannot be true. Vaccine protection is based in an immunological reaction which takes at least two weeks. So, an immediate protective effect against non-COVID-death is not possible. The phenomenon can only be explained by bias, i.e., the vaccine recipients having an inherent better health and decreased risk of death compared with the unvaccinated population, even after matching. Assuming that vaccine effects are specific (since so-called "non-specific effects of vaccines" can be found in observational studies only (please see below)) against the target disease (here COVID-19 associated hospitalization and death) such finding of non-specific immediate beneficial "effects" clearly demonstrate that healthy-wealthy-vaccinée bias is present and that the matching procedure does not at all help. The comparison groups are not comparable when it comes to risk of death. For confirmation of my point, please see Supplemental Tables 11 and 12.

2. Supplemental Figure 11.

Here, COVID-19 vaccine booster is found to be associated with both protection against all-cause hospitalization (A), which is unlikely, and an immediate protection against non-COVID-19 death (B), which cannot be true. Again, restricted matching as suggested by the authors does

not help at all. The comparison groups are not comparable when it comes to risk of hospitalization, and/or death, meaning that the “vaccine effect” is overestimated throughout the study. These observations demonstrate our inability to control for bias in observational studies and must be taken seriously. These observations have nothing to do with vaccine “effects” but are caused by inherent differences between the comparison groups. Of course, the COVID-19 vaccine decreases the risk of severe COVID-19; this was demonstrated in randomised controlled trials. However, it is clearly not easy to validly estimate the protective effect in an observational study like the present.

ANSWER: Thank you. We agree with the reviewer comments and we reinforce the reviewer's argument that estimation of protective effects in observational study is challenging. Additionally, we would like to highlight our serious take on evaluating bias in the vaccine effectiveness field, in this manuscript and in other clinical and methodological articles from our group. As commented by the reviewer, we tried to minimize the potential healthy-vaccine-bias by matching new cohorts on previous influenza vaccine receipt (a possible surrogate for healthy-seeking bias) and excluding frail patients (patients hospitalized a month prior to vaccination); however, although we observed some decrease in bias for all-cause hospitalizations, the observed lower hazards of non-COVID death remained for the vaccination group. This healthy-vaccine bias was observed in previously published population studies (1-2), and available literature has been unsuccessful in demonstrating successful ways of adjusting for this. On the other hand, providing clear estimates of non-COVID outcomes and presenting analysis trying to ameliorate it, as we did even if unsuccessful, help advance the current knowledge.

We updated our discussion section in order to better acknowledge this limitation.

Further adjustment for previous influenza receipt and exclusion of recently hospitalized patients, variables that would capture part of the healthy and health-seeking biases, improved all cause-hospitalization differences between groups, but not non-COVID death. We hypothesize two other reasons for this finding. First, the principal difference between vaccinated and control ones occurred in the immediate period after the vaccine (Suppl. Table 5 and 6), a finding observed by other studies and improbable to be related to the biological action of vaccines. This is likely because the most ill or frail individuals ended-up dying early on after matching, and in a clinical trial they would likely not be eligible to be randomized. Second, part of the non-COVID-19 deaths could be actually COVID-19 deaths, leading to misclassification of the outcome. In the analyzed population, there is a high proportion of non-COVID-19 deaths that occurred without hospitalization (60% for Cohort A and 71% for Cohort B), which associated with a great network of home and palliative care in the region, makes us hypothesize that these individuals are unlikely to get tested (and consequently being diagnosed). ~~Third, indirect effects of COVID-19 vaccines on non-COVID-19 outcomes, still an open field of research. The protection against non-COVID-19 deaths has been observed even when analyzing clinical trials of adenovirus COVID-19 vaccines.~~ (Discussion section, pp 8-9)

'The main limitation of this study is its observational design. Although large observational databases with adequate methodology may duplicate the results of randomized clinical trials'³⁷,

residual confounding cannot be excluded. However, we tried to minimize the chances of confounding by including variables associated with health-seeking behaviors (number of outpatients visits) and socioeconomic factors (the MEDEA deprivation index) in addition to calibrating for negative control outcomes. Visual inspection of bias indicators (i.e., during the initial days following vaccination)^{51,52} showed a low risk of bias and negative control adjustment showed similar results. However, the decreased hazards of non-COVID death, particularly in the immediate period post-vaccination, indicates residual healthy bias. We attempt to reduce this bias by matching new cohorts on previous influenza vaccine receipt and excluding frail patients (patients hospitalized a month prior) and adjusting of variables that could capture healthy-seeking behavior, but lower hazards for non-COVID death persisted and our results should be interpreted in light of these findings. It is possible that the VE and rVE estimates are overestimated because of this residual healthy bias, particularly in the immediate period post-vaccination when most non-COVID deaths were observed. (Discussion section, pp 9-10)

References:

1. William J. Hulme et al., “Challenges in Estimating the Effectiveness of COVID-19 Vaccination Using Observational Data,” *Annals of Internal Medicine* 176, no. 5 (May 2023): 685–93, <https://doi.org/10.7326/M21-4269>.
2. Ori Magen et al., “Fourth Dose of BNT162b2 mRNA Covid-19 Vaccine in a Nationwide Setting,” *New England Journal of Medicine* 386, no. 17 (April 28, 2022): 1603–14, <https://doi.org/10.1056/NEJMoa2201688>.

Please note that despite being persuaded intensively for more than 40 years, the hypothesis the authors refer to about vaccines potentially having “non-specific” effects cannot be confirmed by randomised trial data and is most likely based on – bias in observational studies.

ANSWER: As suggested by the reviewer, we excluded this sentence from the discussion section.

'Second, part of the non-COVID-19 deaths could be actually COVID-19 deaths, leading to misclassification of the outcome. In the analyzed population, there is a high proportion of non-COVID-19 deaths that occurred without hospitalization (60% for Cohort A and 71% for Cohort B), which associated with a great network of home and palliative care in the region, makes us hypothesize that these individuals are unlikely to get tested (and consequently being diagnosed). Third, indirect effects of COVID-19 vaccines on non-COVID-19 outcomes, still an open field of research. The protection against non-COVID-19 deaths has been observed even when analyzing clinical trials of adenovirus COVID-19 vaccines.' (Discussion section, pp 8)

3. The authors wish to stick to the composite outcome of COVID-19-hospitalisation and -death. I do not agree. First, the two outcomes are clearly different in frequency – please see Figure 2. Second, as demonstrated by the supplemental Figures 10-11, the bias differs by outcome. So, these two outcomes cannot be lumped together. As suggested in my first review, the two outcomes must be presented separate only.

ANSWER: Thank you. As suggested by the reviewer and appointed by the editor, we have removed the combined outcome assessment (COVID-19 hospitalization and/or COVID-19 death), and separated both outcomes in all analyses. We defined COVID-19 hospitalization as our new primary outcome, and maintained COVID-19 death as a secondary outcome, due to the low number of events and low statistical power. The change in our primary outcome was followed by an update in all calculations with very minor changes in comparison to our previous results (See *Figures 1, 2, 3, Suppl. Figures 6, 7, 10, 11, 12, 13, Suppl. Tables 3, 4, 13*). The results section was changed accordingly.

In summary, due to significant healthy-and-wealthy vaccinee bias the present study generally overestimate the COVID-19 vaccine protective effect. This well-known problem has not been properly addressed in the revised manuscript.

ANSWER: Thank you. We tried to minimize this bias at a maximum, including sensitivity analyses, recalibrating for negative control outcomes and evaluating non-COVID-19 hard outcomes. We also discussed the limitations associated with the bias.

'The main limitation of this study is its observational design. Although large observational databases with adequate methodology may duplicate the results of randomized clinical trials³⁷, residual confounding cannot be excluded. However, we tried to minimize the chances of confounding by including variables associated with health-seeking behaviors (number of outpatients visits) and socioeconomic factors (the MEDEA deprivation index) in addition to calibrating for negative control outcomes. Visual inspection of bias indicators (i.e., during the initial days following vaccination)^{51,52} showed a low risk of bias and negative control adjustment showed similar results. However, the decreased hazards of non-COVID death, particularly in the immediate period post-vaccination, indicates residual healthy bias. We attempt to reduce this bias by matching new cohorts on previous influenza vaccine receipt and excluding frail patients (patients hospitalized a month prior) and adjusting of variables that could capture healthy-seeking behavior, but lower hazards for non-COVID death persisted and our results should be interpreted in light of these findings. It is possible that the VE and rVE estimates are overestimated because of this residual healthy bias, particularly in the immediate period post-vaccination when most non-COVID deaths were observed. (Discussion section, pp 9-10)

REVIEWERS' COMMENTS

Reviewer #3 (Remarks to the Author):

Thank you for allowing me to re-re-review this interesting and relevant observational study on the effectiveness of COVID-19 vaccines against COVID-19 hospitalization. The majority of my concerns have been addressed, and the revised manuscript has significantly improved.

I have one remaining concern regarding the choice of negative control outcomes, which appears unusual to me. The formal definition of a negative control outcome is one that shares potential sources of bias with the primary outcome but cannot plausibly be related to the treatment of interest. Therefore, in this study, suitable negative control outcomes should reflect general health signals and include measures of health-seeking behavior, such as all-cause hospitalization, non-COVID-19 deaths, and outcomes from Supplemental Table 8, where clear signs of residual confounding are observed.

The outcomes designated as negative control outcomes in Supplemental Figure 8 either represent rare events or are irrelevant due to no overlapping potential sources of bias. While I acknowledge that the authors now address the presence of residual confounding in the analysis, the suboptimal selection of negative control outcomes is concerning. The distribution of Hazard Ratios (HRs) in the negative control outcome forest plot (Supplemental Figure 8) suggests the absence of residual confounding, which is misleading.

REVIEWERS COMMENTS:

Thank you for allowing me to re-re-review this interesting and relevant observational study on the effectiveness of COVID-19 vaccines against COVID-19 hospitalization. The majority of my concerns have been addressed, and the revised manuscript has significantly improved.

I have one remaining concern regarding the choice of negative control outcomes, which appears unusual to me. The formal definition of a negative control outcome is one that shares potential sources of bias with the primary outcome but cannot plausibly be related to the treatment of interest. Therefore, in this study, suitable negative control outcomes should reflect general health signals and include measures of health-seeking behavior, such as all-cause hospitalization, non-COVID-19 deaths, and outcomes from Supplemental Table 8, where clear signs of residual confounding are observed.

The outcomes designated as negative control outcomes in Supplemental Figure 8 either represent rare events or are irrelevant due to no overlapping potential sources of bias. While I acknowledge that the authors now address the presence of residual confounding in the analysis, the suboptimal selection of negative control outcomes is concerning. The distribution of Hazard Ratios (HRs) in the negative control outcome forest plot (Supplemental Figure 8) suggests the absence of residual confounding, which is misleading.

ANSWER: Thank you for the commentary. We understand the reviewer's concern and the willingness to include more general health-seeking behaviors such as all-cause hospitalizations and particularly non-COVID deaths in the negative controls set. However, the choice of negative control outcomes should follow a strict assumption of non-causality between the exposure of interest (ie, vaccine exposure) and the negative outcome (ie, non-COVID death and all-cause hospitalization). We believe we might not have enough evidence to rule out some causal association in the case of proposed negative outcomes. For instance, we have indirect evidence by our data that very likely part of the non-COVID-19 deaths are misclassified and might constitute actually COVID-19 deaths. In our study, we have shown that patients that died of COVID-19 were two times more likely to be hospitalized before death compared to non-COVID deaths (67% vs 30% for Cohort A and 52% vs 39% for Cohort B), a finding that raises the hypothesis that some non COVID-19 deaths were possibly non-diagnosed COVID-19 deaths. At the same time, COVID-19 vaccines could trigger previous chronic conditions exacerbation and be associated with non-COVID-19 hospitalizations, etc. Vaccination effects on these negative outcomes are partial and likely relatively small, as expected, but in any case does not fulfill the non-causal assumption.

In addition, our choice of negative outcomes selection was based on previous validated literature (1), and encompassed a wide range of logically implausible outcomes, which, although more rare, more strictly follows the negative outcomes assumption. Previous studies of similar exposures (influenza vaccines) have chosen a similar structure of negative outcomes (2). Additionally, in our first revision version of this article, as suggested by the reviewer, we have performed an additional analysis including an expanded set of negative outcomes with very similar results. We now update this analysis to our new primary outcome of COVID-19

hospitalization, with similar results (*see below*). We chose to keep our initial selection because of the previous validation of these outcomes (1).

Finally, we would like to reinforce that we believe there might still be residual unmeasured confounding, particularly healthy-vaccine bias, as shown by the magnitude of non-COVID-19 death protection, particularly in the initial days following vaccination. We present all curves in the supplementary material and main results in the results section. We also discussed this limitation in the discussion section. We have now updated our limitations section with a sentence regarding the negative outcomes selection. Additionally, we present the ‘expanded set’ of negative outcomes results in Suppl. Table 4

‘Visual inspection of bias indicators (i.e., during the initial days following vaccination)51,52 showed a low risk of bias and negative control adjustment showed similar results. Although the selection of negative outcomes might be debatable, we have chosen negative outcomes previously validated in very similar settings.’ (Discussion Section)

References:

1. Català M, Burn E, Rathod-Mistry T, Xie J, Delmestri A, Prieto-Alhambra D, et al. Observational methods for COVID-19 vaccine effectiveness research: an empirical evaluation and target trial emulation. *International Journal of Epidemiology*. 2024 Feb 1;53(1):dyad138.
2. Izurieta HS, Lu M, Kelman J, Lu Y, Lindaas A, Loc J, et al. Comparative Effectiveness of Influenza Vaccines Among US Medicare Beneficiaries Ages 65 Years and Older During the 2019–2020 Season. *Clinical Infectious Diseases*. 2021 Dec 6;73(11):e4251–9.

Table (Suppl Table 4): Original results and Calibrated Outcomes with Two Different Sets of Negative Outcomes (Original NCO Outcomes N = 43 and Expanded NCO Outcomes).

Outcome: COVID-19 Hospitalization

Vaccination Status	Original Calibrated Outcomes	Expanded* Calibrated Outcomes
14 days after 1st Dose	42.7% (11.2 - 63.0)	43.9% (15.9 - 62.6)
7 days after 2nd dose	58.2% (43.8 - 68.9)	58.8% (42.6 - 70.5)
14 - 60 days after 3rd dose	73.5% (61.3 - 81.8)	72.1% (59.3 - 80.8)
60 or more days after 3rd dose	50.8% (-1.2 - 76.0)	49.7% (2.1 - 74.2)

* = plus low back pain, mild depression, urinalysis, mammography, throat irritation, chest pain, irregular periods, migraine, acetaminophen, amoxicillin and ibuprofen